# Topoisomerase VI is a chirally-selective, preferential DNA decatenase

**Shannon J McKie[1,2], Parth Rakesh Desai[1], Yeonee Seol[1], Adam MB Allen[2], Anthony Maxwell[2]\*, Keir C Neuman[1]\***

[1]Laboratory of Single Molecule Biophysics, National Heart, Lung and Blood Institute, National Institutes of Health, Bethesda, United States; [2]Department of Biochemistry and Metabolism, John Innes Centre, Norwich, United Kingdom

**Abstract** DNA topoisomerase VI (topo VI) is a type IIB DNA topoisomerase found predominantly in archaea and some bacteria, but also in plants and algae. Since its discovery, topo VI has been proposed to be a DNA decatenase; however, robust evidence and a mechanism for its preferential decatenation activity was lacking. Using single-molecule magnetic tweezers measurements and supporting ensemble biochemistry, we demonstrate that *Methanosarcina mazei* topo VI preferentially unlinks, or decatenates DNA crossings, in comparison to relaxing supercoils, through a preference for certain DNA crossing geometries. In addition, topo VI demonstrates a significant increase in ATPase activity, DNA binding and rate of strand passage, with increasing DNA writhe, providing further evidence that topo VI is a DNA crossing sensor. Our study strongly suggests that topo VI has evolved an intrinsic preference for the unknotting and decatenation of interlinked chromosomes by sensing and preferentially unlinking DNA crossings with geometries close to 90°.

**\*For correspondence:**
tony.maxwell@jic.ac.uk (AM);
neumankc@nhlbi.nih.gov (KCN)

**Competing interest:** The authors declare that no competing interests exist.

## Editor's evaluation

The present work is noteworthy for explaining how DNA topoisomerase VI, an archaeal and plant based enzyme with homology to the Spo11 meiotic recombination core complex, senses DNA crossovers to preferentially remove positive supercoils and DNA catenanes. The findings are important for understanding how topoisomerase VI supports DNA replication and chromosome disentanglement.

## Introduction

The DNA topoisomerases (topos), which are fundamental to cellular survival through the maintenance of genome integrity, manipulate DNA topology via the transient cleavage of the DNA backbone (*Bush et al., 2015*; *Wang, 1996*). This mechanism constitutes a highly vulnerable situation for the duplex; however, topos have evolved exquisite control over this reaction in order to ensure that DNA is cleaved under specific circumstances, then rapidly resealed. There are two topo families: the type I topos, which utilise a transient single-stranded DNA break (SSB), and the type II topos, which utilise a transient double-stranded DNA break (DSB) (*McKie et al., 2021*). These enzymes are vital during DNA metabolism, particularly in the relief of torsional stress built up ahead of and behind the transcription complexes and replication forks, as well as the removal of catenanes and knots (*Bush et al., 2018*; *McKie et al., 2021*; *Pommier et al., 2016*). For this reason, targeting the topos as a means to treat bacterial infections and cancer has had significant and ongoing clinical success (*Bush et al., 2020*; *Cuya et al., 2017*; *Hiasa, 2018*; *Pommier, 2013*).

The type II topos are further subcategorised as type IIA and type IIB (*Gadelle et al., 2003*). This separation is based on structural and evolutionary premises; however, the general type II topo

reaction, known as strand passage, is believed to be shared amongst all type II topos. Strand passage involves the binding of one DNA duplex, termed the gate segment (G-segment), which is transiently cleaved and opened to allow the passage of a second DNA duplex, the transported segment (T-segment), through the break, thereby changing the topological state of the DNA (*Berger et al., 1996*; *Roca et al., 1996*). This reaction is integral to DNA topology maintenance through, but not restricted to, the relaxation of positive supercoils generated ahead of separated DNA strands, and the decatenation and unknotting of replicated genomic material. Even though all type II topos have been demonstrated to be active to some extent in both reactions in vitro (*Bush et al., 2015*), topos seemingly adopt preferential activities in vivo (*McKie et al., 2021*). *Escherichia coli* DNA gyrase (gyrase) has been shown to be an integral component of the replication and transcriptional machinery (*Ahmed et al., 2017*; *Kreuzer and Cozzarelli, 1979*; *Stracy et al., 2019*), relieving torsional strain caused by positive supercoiling and allowing fork progression (*Khodursky et al., 2000*). *E. coli* topoisomerase IV (topo IV), however, is indispensable for the decatenation of the replicated bacterial genome (*Kato et al., 1990*; *Kato et al., 1988*; *Wang et al., 2008*), despite the highly processive and efficient relaxation of positive supercoils in vitro (*Neuman et al., 2009*; *Stone et al., 2003*) and evidence that it can, to an extent, support replication and transcription fork progression in vivo using cells encoding temperature-sensitive gyrase mutations (*Khodursky et al., 2000*). The preferential decatenation activity of topo IV in vivo is thought to be a consequence of protein-protein recruitment via the *E. coil* SMC (*s*tructural *m*aintenance of *c*hromosome) complex, MukBEF (*Hayama and Marians, 2010*; *Li et al., 2010*; *Nicolas et al., 2014*; *Nolivos et al., 2016*), and temporal regulation (*Espeli et al., 2003*).

DNA topoisomerase VI (topo VI) is a heterotetrameric, ATP/$Mg^{2+}$-dependent type IIB topo formed from two Top6A and two Top6B subunits, initially isolated from the hyperthermophillic archaeon, *Sulfolobus shibatae* in 1994 (*Bergerat et al., 1994*). It was later demonstrated to be present throughout the archaeal domain, in some bacteria, and in certain eukaryotes, including plants and algae (*Gadelle et al., 2003*). The Top6A subunit is highly homologous to the eukaryotic meiotic factor, Spo11, crucial for the generation of DSBs during recombination (*Claeys Bouuaert et al., 2021*; *Nichols et al., 1999*). Top6B structural homologues have also been identified in higher eukaryotes, including mouse and *A. thaliana*, and were shown to interact with Spo11 (*Robert et al., 2016*; *Vrielynck et al., 2016*). Recently, the purification of the Spo11 core complex was achieved, and structural data indicated a high degree of similarity to the topo VI heterotetramer (*Claeys Bouuaert et al., 2021*). Therefore, despite the archaeal origins of topo VI, its characterisation has been of significant relevance to eukaryotic genome metabolism, particularly in the study of meiosis, and has also shed further light on the evolution of eukaryotic cells as archaeal descendants (*Gadelle et al., 2003*). Since the discovery of topo VI, its physiological role in the archaea and eukaryotes in which it arises has been unclear. *S. shibatae* topo VI was shown to relax positive and negative supercoils, but decatenated more efficiently, with threefold less enzyme required to decatenate 0.4 µg of kinetoplast (k)DNA than was required to relax 0.4 µg of negatively supercoiled pTZ18 (*Bergerat et al., 1994*). It is worth noting, however, that the number of strand passage events required to decatenate kDNA is not necessarily equal to the number required to relax the same amount of plasmid DNA. Nevertheless, this result, along with the discovery that *A. thaliana* topo VI was crucial during endoreduplication (*Hartung et al., 2002*; *Sugimoto-Shirasu et al., 2002*), a mechanism by which plant cells increase in size through multiple rounds of genome replication in the absence of mitosis (*Sugimoto-Shirasu and Roberts, 2003*), led to the hypothesis that topo VI was likely a decatenase, unlinking replicated genomic material in vivo. However, it has also been speculated that topo VI may be involved in positive supercoil relaxation. Exploring the phylogeny of archaeal topos revealed that archaea also encode topo III, known for its role in decatenation and DNA repair, and that in Euryarchaeota and Crenarchaeota, topo VI was the only encoded topo capable of relaxing positive supercoils (*Forterre and Gadelle, 2009*). This suggested that topo VI must play a role in the removal of transcription- and replication-induced supercoils, particularly in those organisms lacking a DNA gyrase.

Recent structural and biochemical studies have shed light on the mechanism of topo VI (*Corbett et al., 2007*; *Graille et al., 2008*; *Wendorff and Berger, 2018*). Both *Methanosarcina mazei* (*Corbett et al., 2007*) and S. *shibatae* (*Graille et al., 2008*) full length topo VI have been characterised crystallographically, revealing an open and closed conformation, respectively, of a protein with a clamp-like structure forming an interior space large enough to accommodate two DNA duplexes. Top6A harbours the catalytic tyrosine and TOPRIM domain, the dimer of which constitutes a groove of the

correct dimensions to bind the G-segment (**Nichols et al., 1999**). Top6B extends from the Top6A dimer to form a cavity that captures the T-segment, and was demonstrated to be extremely important for tightly coupling ATPase activity to strand passage, indicating that topo VI senses and binds to DNA crossings and bends DNA (**Wendorff and Berger, 2018**). This work also showed that *M. mazei* topo VI was an extremely slow and highly distributive supercoil relaxase. They calculated that at the rate measured using in vitro approaches, it is unlikely that the enzyme functions fast enough to support replication and transcription within the host organism. Therefore, there are still many unanswered questions surrounding topo VI, including its physiological functions, why topo VI specifically is required during endoreduplication in plants, and why the supercoil relaxation activity of topo VI in vitro seems unable to support DNA metabolism in vivo.

Here, using topo VI from *M. mazei* as a model, a single-molecule magnetic tweezers technique was employed, in combination with ensemble biochemistry, to gain a deeper insight into the topo VI mechanism, particularly how DNA geometry modulates activity. Previous research, while instructive, has left us with the conundrum that the in vitro activity of topo VI suggests it would be unlikely to support the cell in terms of transcription- and replication-induced supercoil relaxation (**Wendorff and Berger, 2018**). Here, we present a compelling case for the preferential decatenation activity of topo VI as a structural quality of the enzyme, arising from a strong DNA crossing angle preference of 87.5°, which in turn significantly disfavours the removal of supercoils. We further demonstrate that topo VI also behaves as a DNA crossing sensor, with a dramatic increase in ATPase activity, DNA binding and rate of strand passage, with increasing DNA writhe. Moreover, these data provide a potential explanation for why topo VI remains a vital component of some eukaryotic systems, such as during endoreduplication in plants (**Hartung et al., 2002**; **Sugimoto-Shirasu et al., 2002**), as a dedicated decatenase, the activity of which will not be dominated by the necessity to relax transcription- or replication-induced supercoiling.

## Results

### Topo VI is a slow, chirally selective and highly distributive DNA relaxase

To begin exploring the single-molecule behaviour of *M. mazei* topo VI, a magnetic tweezers supercoiling assay was employed, described in detail elsewhere (**Seol and Neuman, 2011a**; **Seol and Neuman, 2011b**). As opposed to agarose gel analysis of DNA topology, the magnetic tweezers facilitates control over the precise level and chirality of DNA supercoiling, as well as the real-time detection of supercoil relaxation by topo VI with the ability to capture single strand-passage events. At low force (0.2–0.5 pN) right-handed rotation of the magnetic bead positively supercoils the DNA, forming positive writhe (left-handed crossings), and left-handed rotation negatively supercoils the DNA, forming negative writhe (right-handed crossings), both causing the DNA extension to decrease as the plectoneme is formed and extended (**Figure 1A**). At high force, negative supercoiling causes DNA melting rather than formation of negative writhe, hence the DNA extension does not change (**Figure 1A**). Upon addition of topo VI, the DNA extension increases in discrete steps as supercoils are relaxed by topo VI. Therefore, the relaxation reaction can be followed via DNA extension changes over time (**Figure 1B**). First, we measured the chirality-dependent relaxation activity of topo VI using a topo VI titration of 0.25–2 nM at 21 °C, under 0.4 pN of force (**Figure 1C**). The observed DNA extension changes revealed that, on average, topo VI relaxed positive supercoils ~2–3 fold faster than negative supercoils. We found that the average relaxation rates for supercoils of either chirality increased as a function of topo VI concentration with the data fitted to a Michaelis Menten-like equation, resulting in an apparent $K_d$ ($K_{d,app}$) threefold lower for the relaxation of positive DNA writhe. In line with the single-molecule relaxation assay results, the preferential relaxation on positive supercoils by topo VI is also supported by an agarose gel based approach (**Figure 1D**), showing that the relaxation of positively and negatively supercoiled pBR322* was completed by 6 min and 15 min, respectively. Inspection of **Figure 1D** suggests that the relaxation of negatively supercoiled DNA was more processive than positively supercoiled DNA, which was relaxed in a highly distributive manner. However, previous results using a plasmid competition assay determined that topo VI relaxed negatively supercoiled DNA in a highly distributive manner (**Wendorff and Berger, 2018**).

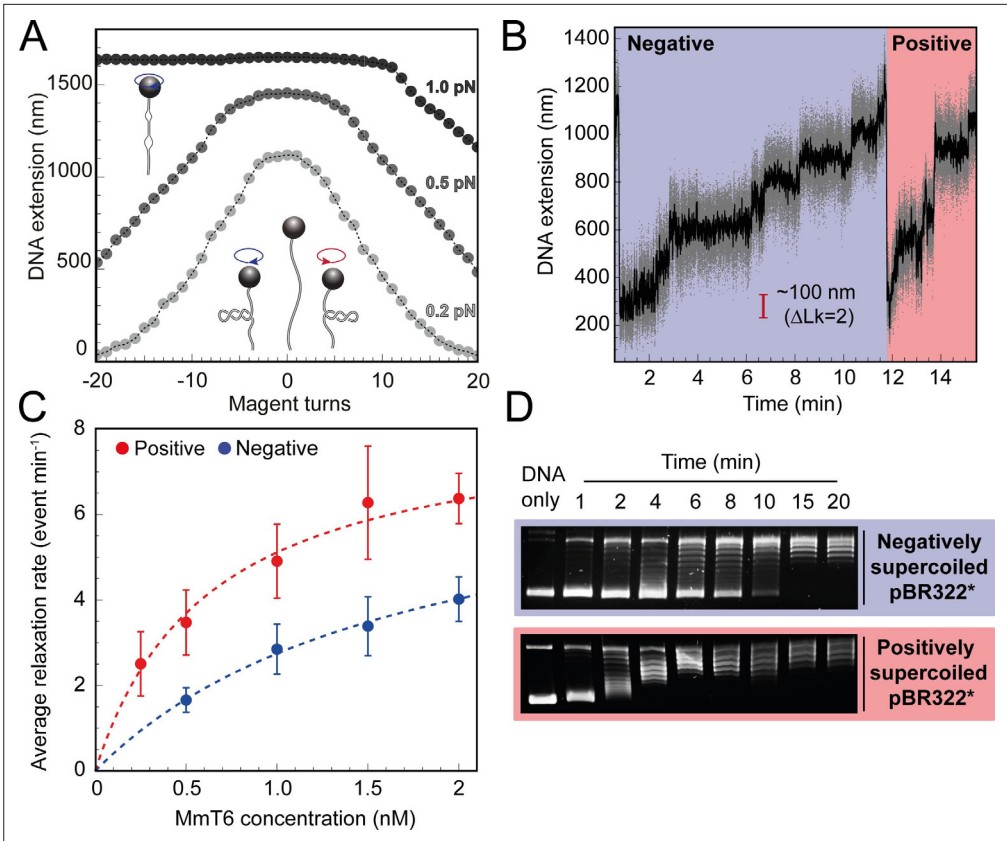

**Figure 1.** Topo VI relaxation rate depends on the chirality of DNA supercoiling. (**A**) Magnetic tweezers calibration curves for a 5 kb DNA duplex supercoiled under low (0.2 pN), medium (0.5 pN) and high (1.0 pN) force. DNA extension is plotted as a function of magnet turns. Negative magnet turn values represent the clockwise rotation of the magnets which produces negative, or right-handed DNA crossings. Positive, or left-handed crossings are produced by rotating the magnets anticlockwise. At high force, clockwise rotation promotes DNA melting, rather than negative supercoiling, hence the DNA extension is insensitive to magnet rotation. (**B**) Example trace of topo VI-dependent supercoil relaxation. Data collected at a force of 0.4 pN, at 21 °C, using 0.5 nM topo VI and 1 mM ATP. Each strand-passage event is evident as an abrupt DNA extension increase of ~100 nm, corresponding to a ΔLk of 2. Relaxation of negative supercoils is highlighted in blue, and positive in red. Positive supercoils are relaxed faster, resulting in short events ( < 1 min) being compressed when plotted on the same axis as negative supercoil relaxation. Data collected at 200 Hz (grey dots) and plotted with a 1 s Savitzky–Golay smoothing filter (black line). (**C**) Average relaxation rate of topo VI (± SEM) on positive (N tethers across all data points = 40) and negative (N tethers across all data points = 42) supercoils as a function of topo VI concentration (0.25–2 nM), collected at a force of 0.4 pN, at 21 °C. Data were fitted to a Michaelis-Menten-like function ($V_0 = \frac{V_{max}[E]}{K_{d,app}+[E]}$). Raw data were analysed in IgorPro 7 (WaveMetrics) using a T-test based method, first described in *Seol et al., 2016*. (**D**) Agarose gel-based supercoil-relaxation time course. Negatively or positively supercoiled pBR322* was incubated at 21 °C, with 20 nM topo VI and the reaction was stopped at consecutive time points using 50 mM EDTA. Samples were run on a 1% (w/v) native agarose gel for 15 hr at ~2 Vcm⁻¹, stained with 0.5 µg/mL ethidium bromide and imaged under UV illumination.

*Figure 1 continued on next page*

*Figure 1 continued*

The online version of this article includes the following source data for figure 1:

**Source data 1.** Source data is in the file *Figure 1*.

For topo VI, each strand-passage event is evident in the trace, as an abrupt DNA extension change of ~100 nm, which is expected for the resolution of a single DNA crossing (*Figure 1B*). This provided preliminary evidence of distributive activity, as many seconds to minutes elapsed between strand-passage events, indicating that topo VI binds and resolves a single crossing before disengaging the G-segment. The analysis was extended to extract each dwell time between strand-passage events and plotting the average as a function of the number of DNA crossings within the plectoneme (*Figure 2*). In the average relaxation rate analysis described above, automatic recoiling of the DNA was initiated 2–3 DNA crossings from full relaxation as it was evident that topo VI relaxation significantly slowed as the plectoneme was relaxed. Using 0.75 nM topo VI, a concentration determined to be effective for both positive and negative plectoneme relaxation, topo VI was allowed to fully relax the DNA (*Figure 2A and B*). This produced characteristic traces in which the dwell times dramatically

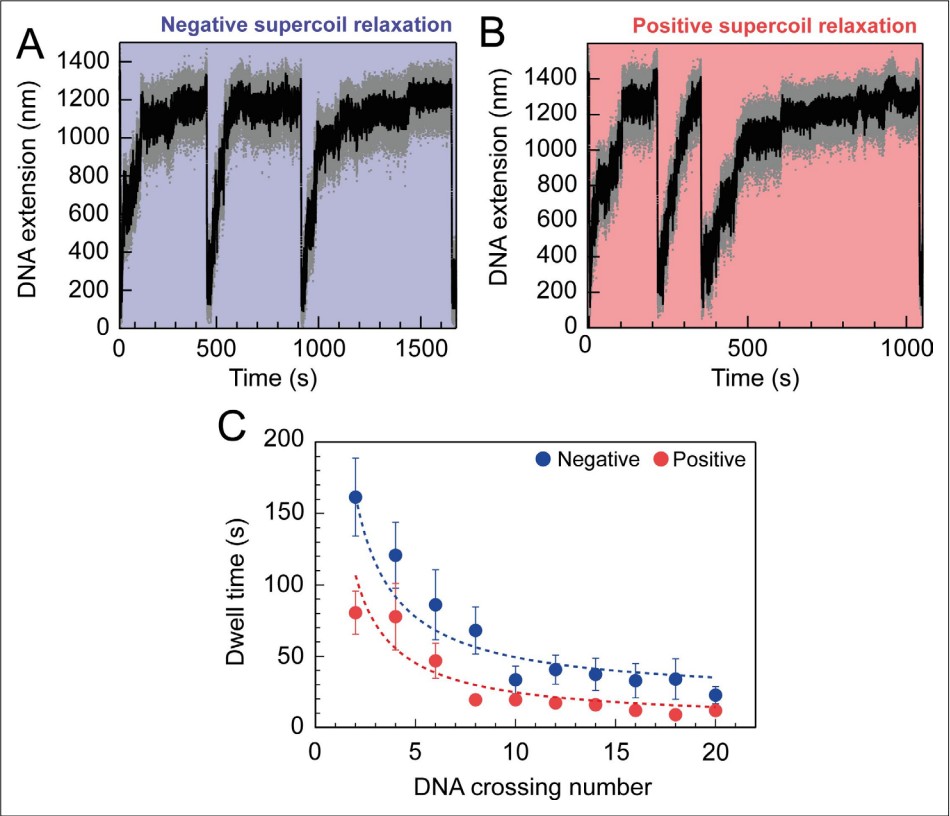

**Figure 2.** Topo VI relaxation rate depend on the level of DNA supercoiling. (**A**) An example trace of 0.75 nM topo VI fully relaxing negative supercoils in a 5 kb DNA duplex at a force of 0.4 pN, at 21 °C. DNA extension is plotted against time. The abrupt decreases in extension correspond to rapid DNA supercoiling by rotating the magnets. The slower DNA extension increases correspond to topo VI-dependent supercoil relaxation activity. Data collected at 200 Hz (grey dots) and with a 1-second Savitzky–Golay smoothing filter (black line)( **B**) same as in A, aside from the DNA being positively supercoiled. (**C**) Dwell time (± SEM) between topo VI-dependent strand-passage events on positive (N dwell times across all data points = 212) and negative (N dwell times across all data points = 146) supercoils, plotted against the level of DNA supercoiling. Data were fitted to an inverse function, where the time taken for topo VI to perform a strand-passage reaction is inversely proportional to the number of DNA crossings present in the substrate. Raw data (**A and B**) were analysed in IgorPro 7 (WaveMentrics) using a T-test based method, first described in *Seol et al., 2016*.

The online version of this article includes the following source data for figure 2:

**Source data 1.** Source data is in the file *Figure 2*.

increased in duration as the DNA approached a fully relaxed state. Plotting these dwell times as a function of the level of DNA supercoiling clearly demonstrated that the reaction rate decreased with decreasing DNA-crossing number. The dwell-time between strand-passage events as a function of plectoneme crossings was well-fitted by an inverse relationship, indicating that the time taken to bind a crossing and perform strand-passage was inversely proportional to the number of DNA crossings present (*Figure 2C*). This finding further supports the distributive nature of topo VI in supercoil relaxation and cements the conclusion described in *Wendorff and Berger, 2018* that topo VI specifically recognises and binds DNA crossings (*Wendorff and Berger, 2018*), making each crossing along the plectoneme a potential site of activity. This specific binding to DNA crossings within the plectoneme is in sharp contrast to topo IV, which relaxed supercoiled DNA independent of the number of crossings, suggesting a single binding site, hypothesised to be the plectoneme end-loop (*Neuman et al., 2009*).

The single-molecule topo VI results are supported by the results of ensemble ATPase and binding assays (*Figure 3* and *Figure 3—figure supplement 1*). Using a radioactive ATPase assay, the rate of ATP hydrolysis was observed to be dramatically stimulated in the presence of negatively-supercoiled pBR322*, in comparison to either relaxed or linearised pBR322*, and tightly coupled to the level of DNA supercoiling, with ATP hydrolysis decreasing once the DNA was relaxed (*Figure 3A*). This was supported by a gel-based relaxation time-course performed in tandem under analogous conditions (*Figure 3B*). With positively-supercoiled DNA, we found that the rate of ATP hydrolysis, measured using a PK/LDH coupled assay, was ~2 fold faster than with negatively-supercoiled DNA (*Figure 3—figure supplement 1*). This difference is consistent with the relaxation rates measured in *Figure 1*. DNA binding by topo VI in the absence of ATP, assayed using a nitrocellulose-membrane capture technique (*Litwin et al., 2015*), also indicates increased topo VI binding with increasing levels of supercoiling (*Figure 3C and D*). However, whereas relaxation and ATPase activity was ~2–3 fold higher for positive writhe than negative, topo VI preferentially bound negatively supercoiled DNA in the absence of ATP. This suggested that the topo VI preference for relaxing positive writhe may be facilitated during a stage post DNA binding, potentially DNA-gate opening or strand passage. Unfortunately, the nitrocellulose-membrane capture technique cannot differentiate between productive and non-productive binding modes, so the results may indicate that more non-productive, such as G-segment-only, DNA binding occurs on negatively supercoiled DNA. However, DNA binding is extremely low in the presence of linearised pBR322*, which suggests G-segment only binding is not the cause of increased binding to negative writhe. Furthermore, the amount of DNA cleavage, as measured using an agarose gel-based assay, also indicates that negatively-supercoiled DNA supports higher levels of ADPNP-dependent cleavage by topo VI than positively-supercoiled DNA (*Figure 3E*). Taken together, these results suggest that topo VI's preference for relaxing positive DNA-crossings may occur at a stage post DNA binding and cleavage, potentially DNA-gate opening and/or strand-passage. Alternatively, it is possible that negatively supercoiled DNA, while bound and cleaved more efficiently, is mildly inhibitory to strand passage. The current data cannot unambiguously differentiate among these possibilities.

A confusing aspect of topo VI activity, previously described in *Wendorff and Berger, 2018* (*Wendorff and Berger, 2018*) and built upon here, is that the supercoil relaxation rate is far slower than the IIA topos. Even at high concentrations (e.g. 2 nM topo VI), the maximum rate measured for positive and negative supercoil relaxation was only 6.4 ± 0.6 and 3.5 ± 0.5 strand-passage events min$^{-1}$, respectively. As detailed in the subsequent section, these rates are 10–50-fold slower than rates measured for the type IIA topos, gyrase and topo IV (*Agarwal and Duderstadt, 2020*; *Basu et al., 2012*; *Stone et al., 2003*). *Wendorff and Berger, 2018* found the maximal topo VI ATP hydrolysis rate, using the PK/LDH assay, to be ~3 ATP min$^{-1}$ during relaxation of negatively supercoiled plasmids (*Wendorff and Berger, 2018*). Here, the ATPase rate was determined to be ~5 ± 0.9 ATP min$^{-1}$, using a radioactive ATPase assay (*Figure 3A*). If 2 ATP molecules are hydrolysed during each strand-passage cycle, both these values correspond well with a rate of ~1.5–3.5 strand passage events min$^{-1}$ on negative supercoils, attained using the magnetic tweezers, between 0.25–2 nM topo VI (3–7 ATP hydrolysed min$^{-1}$). As discussed in *Wendorff and Berger, 2018*, this rate is a fraction of typical type IIA topo rates, suggesting that topo VI is unlikely to function efficiently enough to support cellular DNA metabolism as a DNA relaxase in *M. mazei* (*Wendorff and Berger, 2018*).

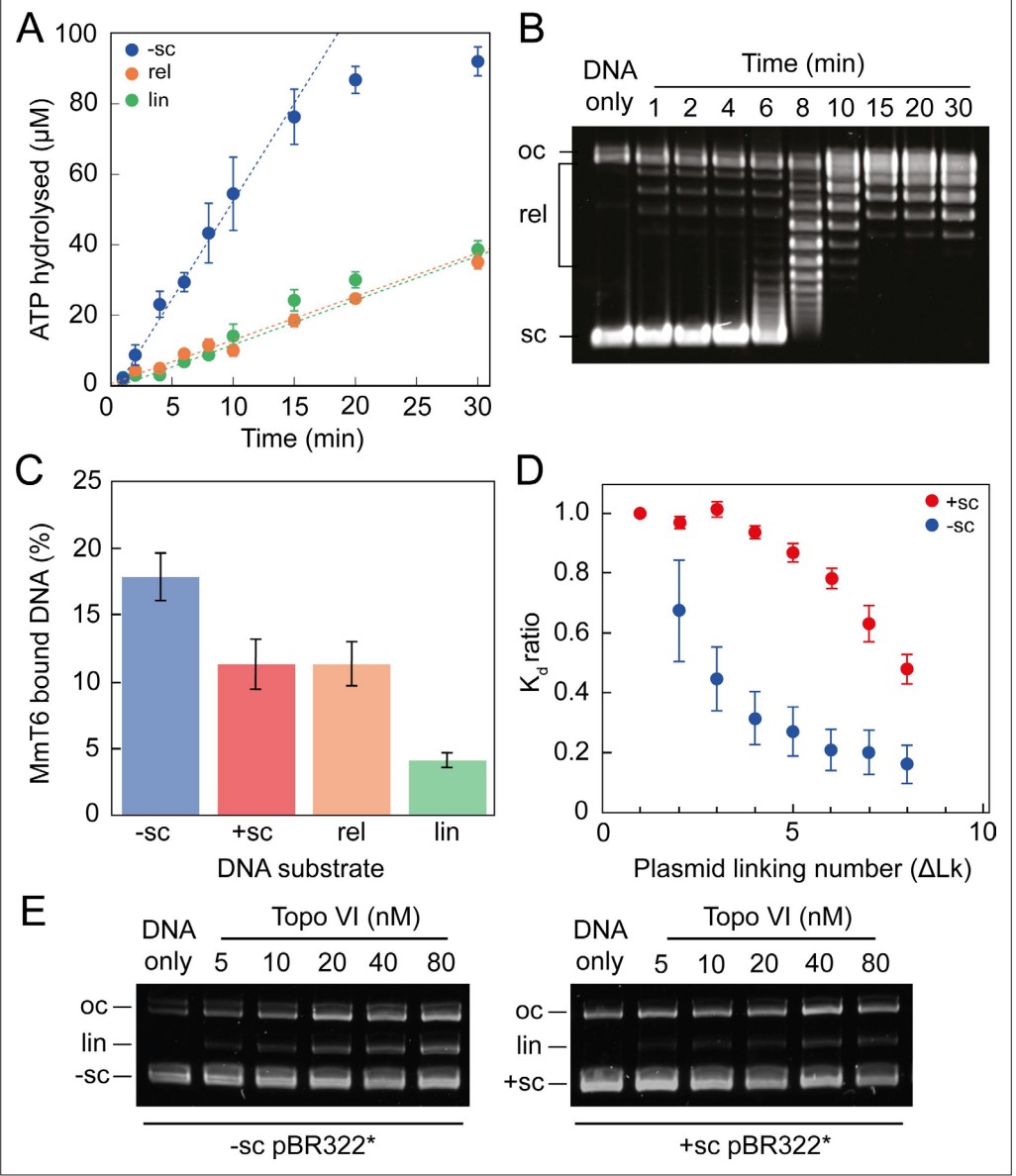

**Figure 3.** ATPase activity and DNA binding of topo VI are stimulated by supercoiled DNA. (**A**) The ATPase activity of topo VI over time, measured using radioactive ATP. Assays were performed at 21 °C, using 1 µM topo VI, 430 nM pBR322* that was negatively-supercoiled (blue), linear (green), or relaxed (orange), and 450 µM [γ-$^{32}$]-ATP. (**B**) Agarose-gel based relaxation time course performed under the same conditions as A, using the same topo VI:DNA ratio (1:2.3), but with non-radioactive ATP. In A, the ATPase rate on supercoiled DNA plateaus around 15 min, which corresponds to the DNA being fully relaxed by topo VI, as shown in B. Samples were run on a 1% (w/v) native agarose gel for 15 hr at ~2 Vcm$^{-1}$, stained with 0.5 µg/mL ethidium bromide and imaged under UV illumination. (**C**) The ATP-independent DNA binding activity of topo VI, measured using a nitrocellulose membrane capture technique, with either negatively-supercoiled (-sc), positively-supercoiled (+ sc), relaxed (rel) or linearised (lin) pBR322* (± SEM). DNA concentrations measured using A$_{260}$. (**D**) The ATP-independent DNA binding activity of topo VI, on either positively- (red) or negatively-supercoiled (blue) topoisomers of pBR322* measured using a nitrocellulose membrane capture technique. Bound and unbound DNA samples were run on a 1% (w/v) native agarose gel for 15 hr at ~2 Vcm$^{-1}$, stained with 0.5 µg/mL ethidium bromide and imaged under UV illumination. The intensity of the bands were measured using ImageJ and the relative dissociation constants (K$_d$± SEM) for each topoisomer calculated as described in *Litwin et al., 2015*. (**E**) DNA cleavage activity of topo VI using negatively- and positively-supercoiled pBR322*. Topo VI concentration was varied from 5 to 80 nM and incubated with 4 nM pBR322*, 1 mM ADPNP, 10 mM MgCl$_2$, at 37 °C for 30 min. All samples were treated with 1 mg/mL proteinase K

*Figure 3 continued on next page*

*Figure 3 continued*

and 0.2% SDS, then run on a 1% (w/v) native agarose gel for 10 hr at ~2 Vcm$^{-1}$, stained with 0.5 μg/mL ethidium bromide and imaged under UV illumination.

The online version of this article includes the following source data and figure supplement(s) for figure 3:

**Source data 1.** Source data is in the file *Figure 3*.

**Figure supplement 1.** ATPase activity of *M. mazei* topo VI with different DNA species.

**Figure supplement 1—source data 1.** Source data is in the file *Figure 3—figure supplement 1*.

## Topo VI is more active in DNA braid unlinking than supercoil relaxation

As shown in *Figures 1 and 2*, topo VI relaxation activity was highly distributive, independent of the supercoil chirality, yet topo VI preferentially relaxed positive writhe. Interestingly, other type IIA topos including *E. coli* topo IV and human topo IIα also demonstrate preferential relaxation of positive writhe (*Neuman et al., 2009*; *Seol et al., 2013*). The detailed basis for chiral preference varies amongst type II topos, however one commonality was that enzymes preferentially act on a particular DNA crossing geometry for either G-segment binding or T-segment capture. In order to explore how DNA-crossing geometry and twist play a role in chirality sensing by topo VI, we employed a magnetic tweezers-based DNA-braiding assay. In this assay, rather than tethering a single torsionally-constrained duplex, two torsionally-unconstrained DNA duplexes are attached to a single magnetic bead, which, upon rotation of the magnets, were wrapped around one another to create writhe without changing twist (*Charvin et al., 2003*; *Figure 4A*). In contrast to the supercoiled substrate, DNA writhe is created directly in the braiding system (rather than via the conversion of twist to writhe); therefore left-handed magnet rotation forms positive writhe (left-handed DNA crossings) and right-handed rotation forms negative writhe (right-handed DNA crossings). The braided DNA substrate is more akin to catenated rather than supercoiled DNA, and allows the exploration of how writhe affects enzymatic behaviour in the absence of twist. In the case of topo VI, the braided DNA substrate had a surprising effect on activity. In *Figure 4B*, the example trace of braided DNA relaxation by 0.1 nM topo VI (0.5 pN), demonstrates that the braid was relaxed in three rapid bursts, at an average rate of ~0.5 strand-passage event s$^{-1}$. This is an example of a trace where the braid was relaxed very quickly, however, even on average the rate of braid unlinking measured over a range of topo VI concentrations (0.05–0.9 nM) increased ~5 fold above that of supercoil relaxation (*Figure 4C*). For example, 0.5 nM topo VI relaxed positive supercoils at a rate of 3.5 ± 0.8 strand-passage events min$^{-1}$, and positive braids at a rate of 18.9 ± 2.3 strand-passage events min$^{-1}$. Moreover, topo VI exhibited robust unlinking activity at concentrations ~10-fold lower than was achievable in supercoil relaxation, and with a limited processivity, passing consecutive T-segments while remaining bound to the initial G-segment. Rates for topo VI processive activity (dwell times between events not included in the average) approach ~0.8 strand-passage events s$^{-1}$ on average (during positive braid unlinking using 0.9 nM topo VI) (*Figure 4D*), which is ~10 fold higher than the rate of positive supercoil relaxation at comparable topo VI concentrations (~0.08 strand-passage events s$^{-1}$ using 1 nM topo VI). In line with this, when topo VI activities were assayed using a singly-catenated, supercoiled DNA substrate, the decatenation reaction (*Figure 5*, left-hand agarose gel) occurred at topo VI concentrations 10-fold lower than for supercoil relaxation (*Figure 5*, right-hand agarose gel). A similar result was also seen for topo IIα (*Waraich et al., 2020*). This demonstrates that DNA braids and catenanes likely share a common geometry favourable for topo VI activity that is not as prevalent in supercoiled DNA (*Figure 5*). Note that in *Figure 5* (left-hand side) the appearance of supercoiled, decatenated products also occurs at a ~10 fold lower concentration than the fully relaxed, decatenated products. Overall, the rate of braid unlinking by topo VI is on par with those of other type IIA topos on their preferred substrates. For instance, *E. coli* DNA gyrase, measured using a rotor bead tracking technique, showed one strand-passage event s$^{-1}$ (*Basu et al., 2012*), or using magnetic tweezers, 1.26 strand-passage events s$^{-1}$ (*Agarwal and Duderstadt, 2020*), and *E. coli* topo IV was demonstrated to relax positively supercoiled DNA at 2.5 strand-passage events s$^{-1}$ (*Stone et al., 2003*).

The average unlinking rates (including dwell times between short processive bursts of activity) of topo VI at concentrations ranging from 0.05 to 0.9 nM were well described by a Michaelis-Menten-like equation, providing $V_{max}$ values of 21.4 ± 0.5 and 10.9 ± 1.3 strand-passage events min$^{-1}$, and $K_{d,app}$ of 67 ± 7 and 164 ± 72 pM, for the unlinking of positive and negative braids respectively (*Figure 4C*). The

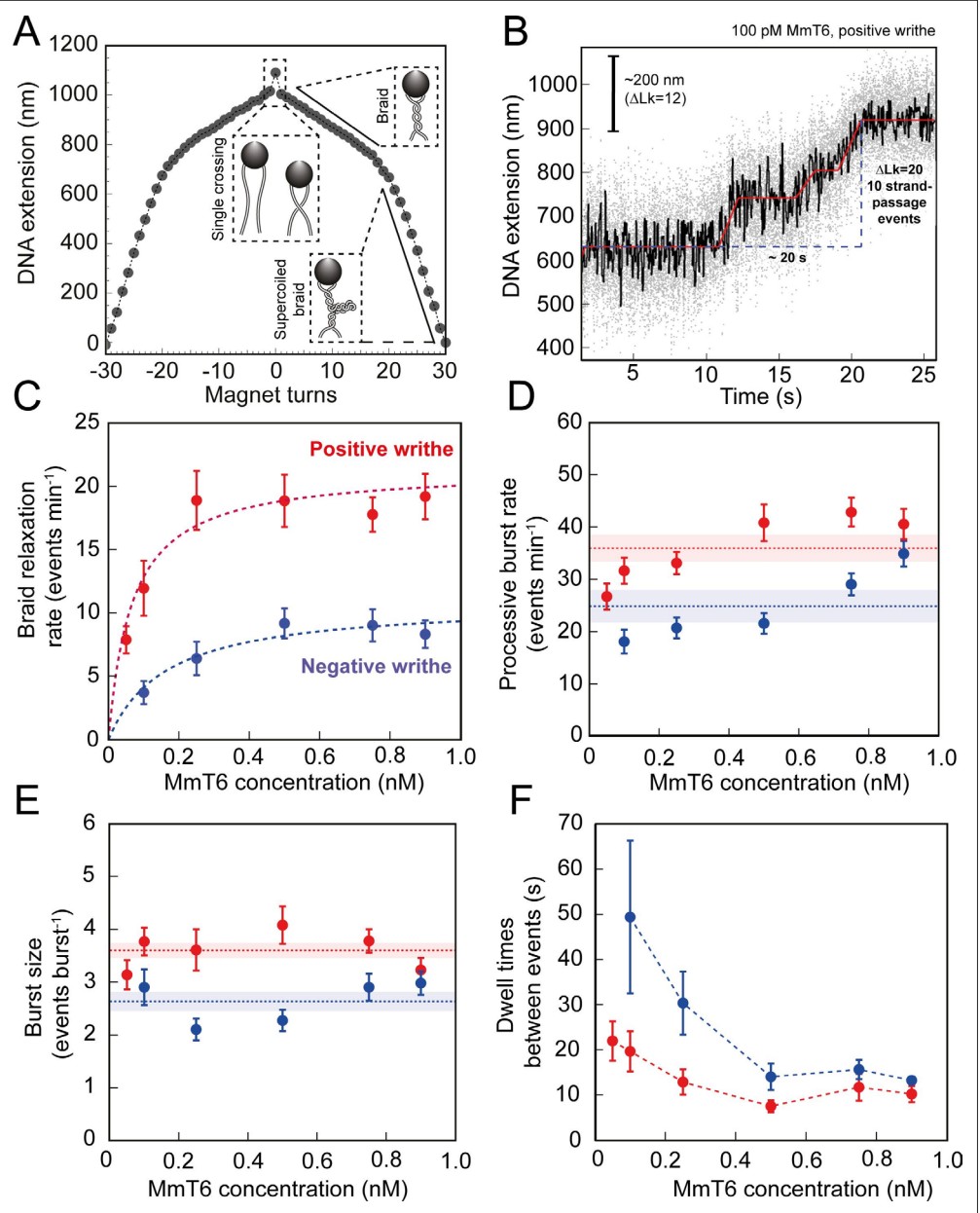

**Figure 4.** Topo VI activity on braided DNA substrates. (**A**) Calibration curve for a DNA braid formed from two 5 kb DNA duplexes tethered to a single magnetic bead. DNA extension is plotted as a function of magnet turns. Negative magnet-turn values represent the right-handed rotation of the magnets producing negative writhe, and positive magnet-turn values represent the generation of positive writhe via left-handed magnet rotation. (Note: this is the reverse scenario of forming a plectoneme, see *Figure 1A*). The first positive or negative 360° turn results in a sharp decrease in DNA extension as a single crossing is input. This is followed by a gradual decrease in extension with rotation, representing the formation of a DNA braid. At a critical number of turns, the braid buckles upon itself to form a supercoiled braid, which is evident in the graph as a switch to a steeper gradient. (**B**) An example of raw magnetic tweezers data, showing topo VI relaxation activity on a DNA braid with positive chirality. Data collected at a force of 0.5 pN, at 21 °C, using 0.1 nM topo VI and 1 mM ATP. Scale bar (black) represents ΔLk of 12, which corresponds to a change in DNA extension of 200 nm. A total of 10 DNA crossings are relaxed by topo VI in ~20 s (blue dashed line), measured as the time between the imposition and complete relaxation of the braids. Data collected at 200 Hz (grey dots) and plotted with a 1 second Savitzky–Golay smoothing filter (black line) and the T-test fit in red (*Seol et al., 2016*). Additional examples of braid relaxation data and the T-test fits are provided in *Figure 4—figure supplement 1*. (**C**) The average rate of topo VI braid unlinking activity (± SEM), of both positive (N tethers across all data points = 92) and negative (N tethers across all data points = 55)

*Figure 4 continued on next page*

*Figure 4 continued*

braids, measured as the number of strand-passage events/min and plotted as a function of topo VI concentration (0.05–0.9 nM). Data were fitted to a Michaelis-Menten-like function ($V_0 = \frac{V_{max}[E]}{K_{d_{app}}+[E]}$). (**D**) The processive burst rate of topo VI (± SEM) on both positive (N burst events across all data points = 206) and negative (N burst events across all data points = 104) braids, measured as the average number of events min$^{-1}$ in a burst, and plotted as a function of topo VI concentration. A burst is defined as rapid topo VI activity corresponding to the passage of two or more consecutive T-segments in which individual strand-passage events cannot be discerned by the step-finder. Any single strand-passage events detected were omitted from the average. The horizontal dashed lines represent the average processive burst rate (± SEM) across all concentrations of topo VI assayed. (**E**) The average burst size of topo VI (± SEM) on both positive (N burst events across all data points = 217) and negative (N burst events across all data points = 132) braids, measured as the average number of strand-passage events per burst, plotted as a function of topo VI concentration. Single passage events were included in the average burst size. The horizontal dashed lines represent the average processive burst size (± SEM) across all concentrations of topo VI assayed. (**F**) The dwell times between processive bursts of topo VI activity on both positive (N dwell times across all data points = 156) and negative (N dwell times across all data points = 119) braids, plotted as a function of topo VI concentration. A dwell time is defined as a period of time in which the DNA extension remains constant, reflecting lack of topo VI-dependent braid unlinking activity. In **C-F**, data was collected at a force of 0.5 pN, at 21 °C, using 1 mM ATP, with topo VI activity on positive DNA braids in red, and in blue for negative DNA braids. Raw data were analysed in IgorPro 7 (WaveMentrics) using a T-test-based method, first described in *Seol et al., 2016*. *Figure 4— figure supplement 2* provides a comparison between the analysis of the experimental braid relaxation data and the analysis of simulated purely distributive braid relaxation data. *Figure 4—figure supplement 3* provides examples of the t-test based fitting of the simulated data sets.

The online version of this article includes the following source data and figure supplement(s) for figure 4:

**Source data 1.** Source data is in the file *Figure 4*.

**Figure supplement 1.** Additional examples of T-test fits to braid relaxation data.

**Figure supplement 1—source data 1.** Source data is in the file *Figure 4—figure supplement 1*.

**Figure supplement 2.** Comparison of the experimental braid relaxation data with a purely distributive relaxation model via simulations.

**Figure supplement 2—source data 1.** Source data is in the file *Figure 4—figure supplement 2*.

**Figure supplement 3.** Examples of T-test based fits to simulated braid relaxation data.

**Figure supplement 3—source data 1.** Source data is in the file *Figure 4—figure supplement 3*.

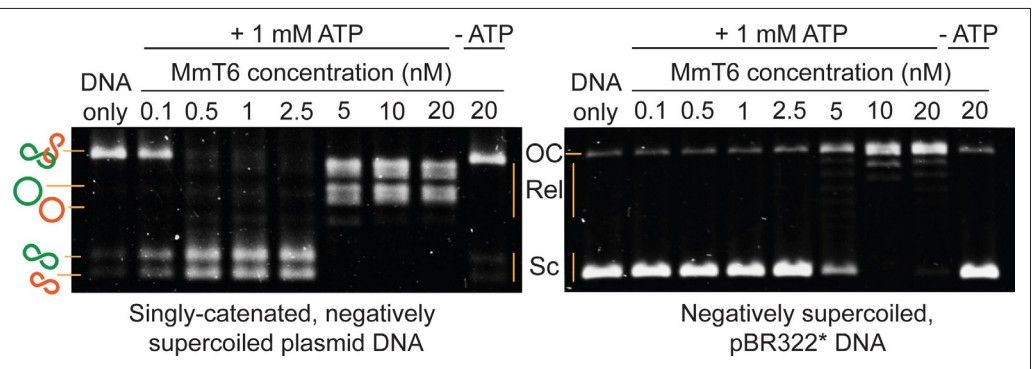

**Figure 5.** Agarose gel-based assay of DNA decatenation and relaxation by *Methanosarcina mazei* topo VI (MmT6). On the left, a singly catenated (depicted by the linked green and orange circles), negatively supercoiled plasmid substrate is decatenated by 0.1–20 nM topo VI. The catenated, supercoiled plasmids vary in size and when decatenated, can be seen as two bands that migrate further (depicted by the separated green and orange circles) (*Waraich et al., 2020*). As they are also negatively supercoiled, the relaxation of the plasmids can be seen at topo VI concentrations ~10 fold higher (5 nM) than when full decatenation is seen (0.5 nM). This is further corroborated by a relaxation assay performed using negatively supercoiled pBR322* (right-hand gel), where relaxation activity is not detected until ~10 fold the MmT6 concentration (5 nM) necessary for decatenation. OC: open circular, Rel: relaxed and Sc: supercoiled. Both reactions were incubated for 30 min at 37 °C.

chiral preference remains, with the $V_{max}$ ~2 fold higher, and the $K_{d,app}$ ~2.5 fold lower, for positive-braid unlinking in comparison with negative. This further suggests that the chiral selection originates from a DNA-crossing-geometry sensitive step that occurs between DNA binding and strand-passage. Based on DNA binding and cleavage experiments (*Figure 3C–E*), the chiral selection potentially occurs after G-segment binding and cleavage as topo VI exhibits tighter binding and higher levels of cleavage in the presence of negative writhe than positive. This suggests that DNA-gate opening and/or strand passage is sensitive to the crossing angle, with a preference for angles more commonly found in positive writhe.

The average rate of unlinking by topo VI (rate including dwell times) was approximately three-fold higher than for supercoil relaxation, largely due to bursts of two or more unlinking events in rapid succession. This rapid unlinking could be due to either simultaneous binding and unlinking by multiple enzymes, or a processive unlinking by a single enzyme, which can be distinguished based on the enzyme concentration dependence. While the processive rate modestly increased over large changes in topo VI concentration (~1.5 fold increase for positive, ~2fold increase for negative) (*Figure 4D*), the size of the processive bursts remained constant (*Figure 4E*) and the duration of the dwell times between events decreased to a minimum (*Figure 4F*). Together these results support the hypothesis that the burst activity is due to processive unlinking by a single enzyme. This conclusion is further supported by a comparison of the experimental braid unlinking results with simulations of perfectly distributive braid relaxation occurring with the same average rate (*Figure 4—figure supplements 2 and 3*). The experimental average step-sizes, probabilities of processive bursts, and average dwell times between relaxation events, are substantially larger

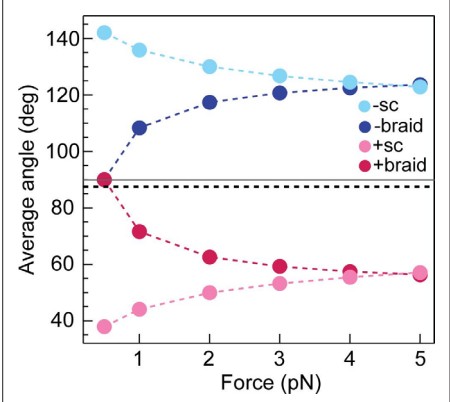

**Figure 6.** Calculated average DNA-crossing angles for supercoils and braids, as a function of force. The average DNA-crossing angle in positive and negative supercoils (+ sc [ink] and -sc [light blue], respectively) were calculated as described in **Neukirch and Marko, 2011**. The temperature was 293 K, the DNA persistence length was 50 nm and the monovalent salt concentration was 100 mM. The average DNA-crossing angles in positive and negative braids (+ braid [red] and -braid [dark blue], respectively) were calculated as described in **Charvin et al., 2005**. The temperature, DNA persistence length and monovalent salt concentration were the same as for supercoils. DNA duplex spacing was 600 nm, DNA-crossing number was eight and the DNA was 5080 bp long. In both the supercoil and braiding calculations, variables were set to mirror experimental conditions as close as possible. The horizontal grey line represents 90° and the dotted black line represents the topo VI DNA-crossing angle preference ($\alpha_0 \approx 87.5°$).

The online version of this article includes the following source data and figure supplement(s) for figure 6:

**Source data 1.** Source data is in the file *Figure 6*.

**Figure supplement 1.** Effect of force on the braid unlinking and supercoil relaxation activity of topo VI.

**Figure supplement 1—source data 1.** Source data is in the file *Figure 6—figure supplement 1*.

than the values from the simulations analysed in with the same t-test step fitting routine (*Figure 4—figure supplements 2 and 3*), consistent with a model in which topo VI relaxes braided DNA in short processive bursts between which the enzyme dissociates. This striking outcome suggests that a unique geometric property more common in DNA braids than supercoils allows topo VI to rapidly catalyse multiple strand-transfer cycles before detaching. DNA binding and ATPase experiments show that topo VI inefficiently binds linear DNA, and the presence of this substrate does not greatly stimulate ATPase activity, indicating that topo VI does not bind well to the G-segment alone. This suggests that during the topo VI strand-passage reaction there is competition between the capture and passage of a T-segment at the correct geometry and the rate of topo VI dissociation from the DNA, with the former likely being accelerated in the case of DNA braids.

Additional experimental data exploring the effects of force on braid unlinking and supercoil relaxation are broadly consistent with a model of topo VI preferentially catalysing strand-passage on two segments juxtaposed at larger crossing angles. Calculations of the average braid and supercoil angles as a function of force, corresponding to the experimental configurations of DNA length, buffer

conditions, and spacing between braided DNA molecules, suggest that DNA crossing-angles in braids were closer to 90° than those in supercoils (*Figure 6*) and that this may be the critical geometric factor promoting DNA unlinking by topo VI. This is particularly true for braids with a larger DNA spacing ( > 500 nm), which favours larger crossing angles (*Charvin et al., 2005*). It is also intriguing that topo VI transitioned from being highly distributive on supercoils to somewhat processive on braids. Positing a crossing angle-dependent strand-passage rate offers a potential explanation for this observation. Specifically, the diffusion of the T-segment into the topo VI cavity appropriately juxtaposed to the G-segment for strand-passage may occur frequently in braids, but rarely in supercoils. Measurements of the supercoil and braid relaxation rate as a function of applied force lend support to this proposal that the DNA-crossing angle may be the key determinant of the strand-passage rate (*Figure 6—figure supplement 1*). With increasing force, the average rate of braid unlinking drops significantly (*Figure 6—figure supplement 1A*), for both positive and negative writhe, mainly through an increase in dwell times between events with increasing force, although there is also a slight decrease in the rate and extent of the processive bursts (*Figure 6—figure supplement 1*). This force-dependent rate reduction could be the result of two different factors. It may be due to a force-dependent step in the catalytic cycle that becomes rate limiting, for example, bending the gate segment (*Hardin et al., 2011*; *Thomson et al., 2014*), or closing the DNA gate against the applied tension. However, it is more likely due to the deviation of the DNA-crossing angle away from the preferred angle, which decreases strand-passage rate. This interpretation is bolstered by the supercoil relaxation rate of topo VI as a function of force; the rate initially increased as the force was increased, but then rapidly decreased at higher forces ( >1 pN for positive and >0.6 pN for negative). These results suggested that two competing factors were affecting the rate of relaxation with increasing force (*Figure 6—figure supplement 1E-F*). Consistent with the calculations (*Figure 6*), the initial increase in rate could be due to the DNA-crossing angle getting closer to 90° with increasing force. The cause of the subsequent decrease in activity, particularly for positive supercoil relaxation above 1 pN (*Figure 6—figure supplement 1F*) is less clear. Potential explanations include inhibition of DNA gate closing, or G-segment bending against high force. Alternatively, an increase in DNA twist, which also increases with increasing force, may inhibit DNA binding or cleavage. Whereas the data is largely consistent with a model in which topo VI requires the T- and G-segments to be juxtaposed at a crossing angle close to 90°, directly measuring the preferred crossing angle would provide definite support for this model.

## Topo VI has a strong preference for DNA-crossing angles slightly below 90°

In order to directly determine topo VI's preferential DNA-crossing geometry, we measured unlinking rates of a single DNA-crossing by topo VI in which the crossing geometry can be well-defined and therefore its effect on activity attained. This method was first described in *Neuman et al., 2009* and applied to *E. coli* topo IV. Here, a single-crossing is defined as the interlink between the two DNA duplexes formed by one full magnet rotation (360°) corresponds to a change in linking number of 2. Shown in *Figure 4A*, when the braid goes from fully relaxed to a single crossing, there is a distinct drop in DNA extension. This allows straightforward measurement of the unlinking rate of a single DNA-crossing of either chirality. These measurements can be conducted at high topo VI concentration, as they are complete in a single catalytic event, so as to ensure T-segment binding, rather than G-segment binding, is rate limiting. This, when combined with Monte Carlo and Brownian dynamics simulation-derived DNA crossing-angle distributions, allowed determination of the topo VI DNA crossing-angle preference. This is facilitated as single-positive and -negative crossings are identical in every respect, aside from the crossing angle distributions. An enzyme with a preference for a crossing angle below 90° will have increased activity on positive crossings, as is true for topo IV, whereas if the enzyme binds preferentially to perfectly symmetric DNA-crossings (90°), there would be no difference between the rate of relaxation on positive and negative crossings, as is seen for yeast topo II (*Neuman et al., 2009*).

To begin, the braid being assayed requires precise calibration of the first crossing, which is then fitted to a geometric function (*Figure 7—figure supplement 1*). This allows the calculation of both the length and spacing of the DNA attached to the bead. This information, along with the magnetic force applied to the bead and temperature, is integral to executing the DNA-crossing angle simulations. Once calibration was complete, data collection in the presence of topo VI was performed, imposing

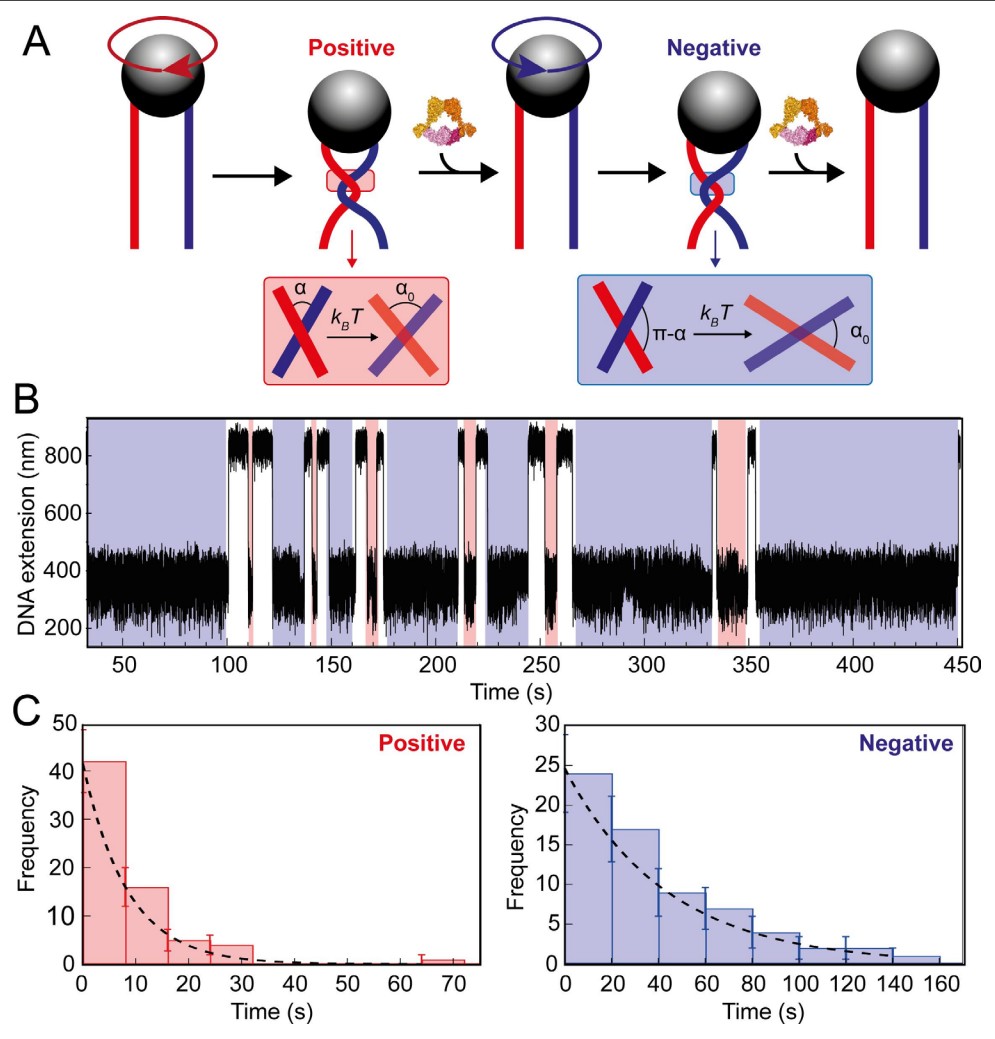

**Figure 7.** Topo VI unlinking single DNA crossings. (**A**) Single crossing assay schematic with DNA crossing geometry for positive (left-handed, red) and negative (right-handed, blue) DNA writhe. One 360° clockwise magnet rotation imparts a positive crossing, which is unlinked by topo VI, followed by the generation and subsequent topo VI-dependent unlinking of a negative DNA crossing, formed by a 360° anticlockwise magnet rotation. The crossing angle is defined as the clockwise angle between the top and bottom DNA strands. For positive crossings this is an acute angle ($\alpha$); for negative crossings the obtuse angle is the supplement ($\pi$-$\alpha$) of the positive angle. For topo VI, achieving the preferred angle ($\alpha_0 < 90$) requires a smaller thermal fluctuation of positive crossings, therefore there is a higher probability of $\alpha_0$ being achieved than for negative crossings. (**B**) Single crossing unlinking data, collected for a braid formed from 3 kb DNA tethers, spaced 624 nm apart, at a force of 1 pN, using 0.9 nM topo VI and 1 mM ATP. Positive crossings (red) were relaxed more rapidly than negative crossings (blue). (**C**) Distributions of the topo VI-dependent unlinking times for negative (blue bars) and positive (red bars) crossings, of the data shown in B. The data were fitted with single exponentials, $P(t) = \tau^{-1}\exp(-t/\tau)$, returning characteristic unlinking times of $\tau_R = 47 \pm 7$ s for negative crossings, and $\tau_L = 8 \pm 1$ s for positive crossings, giving a ratio of $\tau_L/\tau_R = 0.19 \pm 0.04$.

The online version of this article includes the following source data and figure supplement(s) for figure 7:

**Source data 1.** Source data is in the file *Figure 7*.

**Figure supplement 1.** Braid tether calibration and geometric fit.

**Figure supplement 1—source data 1.** Source data is in the file *Figure 7—figure supplement 1*.

a DNA crossing of one chirality, allowing topo VI to unlink the DNA-crossing, before imposing one of the opposite chirality and so on (*Figure 7A and B*). Not all braid geometries were unlinked by topo VI, in particular the more acute crossing angles, determined by assaying the braid for up to an hour without a single event, or until the braid became unattached from the slide surface. The distribution

of times taken to relax either positive (left-handed) or negative (right-handed) crossings were plotted as histograms and fitted with single exponentials to attain the characteristic unlinking times for the DNA-crossing geometry being assayed (*Figure 7C*). In agreement with results from DNA braids and supercoils, the characteristic unlinking time from the fitting on positive crossings ($\tau_L$) for the particular braid assayed in *Figure 6* was 8.4 ± 1.4 s, which was 4.7-fold faster than that of negative crossings ($\tau_R$) (43.8 ± 6.9 s), suggesting that a positive crossing geometry facilitates topo VI activity through the higher probability of forming the preferred crossing angle (*Figure 7C*).

To relate this experimentally-derived data to a more precise value for the DNA-crossing angle preference of topo VI, we performed both Monte Carlo (MC) (*Figure 8A*) and Brownian Dynamics (BD) simulations to determine the distribution of crossing angles formed for a given DNA-crossing geometry and force. The crossing angle distribution is attained by simulating the thermal fluctuations of two DNA molecules, the movements of which are dependent on DNA length, the tension applied to the DNA, temperature, and the spacing between the DNA duplexes. So, even though the average crossing angle is not necessarily the preferred crossing angle ($\alpha_0$), this fluctuation results in $\alpha_0$ being formed at a certain frequency, which is related to the measured unlinking rate of a single DNA-crossing (assuming that achieving $\alpha_0$ is rate-limiting). The MC and BD simulations can predict how probable $\alpha_0$ is under the given DNA-crossing geometry for positive and negative crossings. If the assumptions that the time taken to unlink the crossing is dependent on a single rate-limiting step, which is likely as the unlinking times are exponentially distributed, and that this step is dependent on the DNA-crossing angle, then $\tau_L / \tau_R$ is equal to the ratio of negative and positive DNA-crossing angle probabilities (*Figure 8B*). In other words, for the DNA-crossing geometry shown in *Figures 7 and 8*, the positive crossing is relaxed 4.7-fold faster than the negative, and so, using this ratio, the angle that is 4.7-fold more probable in positive DNA crossings than negative can be determined and defined as $\alpha_0$. This was repeated for fourteen different crossing geometries that all yielded extremely similar values for $\alpha_0$, over a wide range of average crossing angles (*Figure 8C*) and applied forces (*Figure 8D*).

Together these data indicate that the preferred DNA-crossing angle for topo VI is 87.8° ± 0.4°, when using MC simulations, and 87.4° ± 0.4° (uncertainties represent SEM), when using BD simulations. From a technical perspective, simulating the crossing angle distributions using two distinct simulation techniques and attaining strikingly similar values for topo VI $\alpha_0$, not only adds confidence to the accuracy of this value but also supports the use of either simulation technique in measuring DNA-crossing angle distributions. Supported by MC simulations done by *Stone et al., 2003*, the topo VI angle preference can account for the consistent twofold difference in rate between the positive and negative supercoil relaxation and braid unlinking assays. Furthermore, combining the ATPase measurements of topo VI relaxing positively and negatively supercoiled DNA (*Figure 3—figure supplement 1*) with crossing angle distributions of supercoiled plasmid DNA obtained from MC simulations performed by Vologodskii and Cozzarelli (*Vologodskii and Cozzarelli, 1996*; *Vologodskii and Cozzarelli, 1994*) permits an independent estimate of the preferred crossing angle of topo VI. This analysis returns a preferred crossing angle in the range of 83° to 87° (*Figure 8—figure supplement 1*), in excellent agreement with the value of 87.6° obtained from the more precise single molecule measurements.

The preferred crossing angle for topo VI is similar to that found for topo IV, attained using the same single-crossing assay, of 85.5° (*Neuman et al., 2009*). However, one distinct difference is that topo VI exhibited a far stricter preference for the DNA-crossing geometries on which it would act. If the imposed crossing angle ($\alpha_L$) is defined as the average angle in the positive distribution (see *Figure 8A*), then the most acute $\alpha_L$ that topo VI could unlink was 77°, with no activity detected on braids with an $\alpha_L$ lower than this (*Figure 8—figure supplement 2*). However, topo IV seemed far more versatile and was able to unlink DNA-crossings with an $\alpha_L$ as low as 50° (*Neuman et al., 2009*). This suggests that topo VI either cannot remain bound to a disfavoured DNA crossing for long enough to achieve the preferred angle, or can bind but cannot achieve the correct crossing angle for gate opening and/or strand-passage. The latter is supported anecdotally by the observation that on some single crossings with more acute crossing angles, when manual removal of the single crossing was attempted by turning the magnets, the DNA extension did not increase. This suggests topo VI has bound to the crossing but unable to pass the duplex or reopen the N-gate, holding the two duplexes in place.

Taken together, the data obtained from the single-crossing assays not only provides an explanation for chiral discrimination by topo VI, but also strengthens the idea that topo VI is a preferential

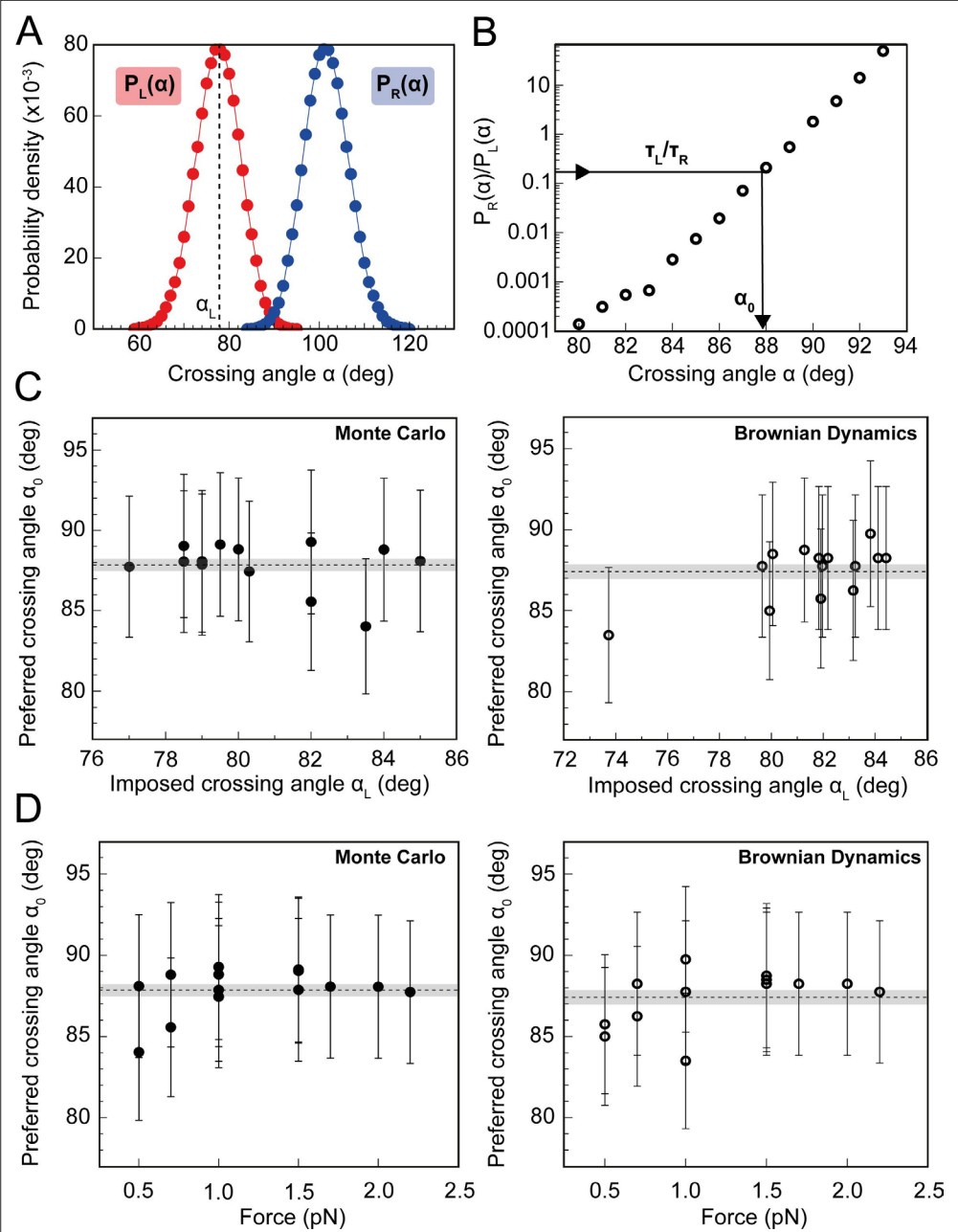

**Figure 8.** Preferred DNA crossing angle measurements for topo VI. (**A**) Crossing angle probability distributions for single positive ($P_L(\alpha)$, red) and negative ($P_R(\alpha)$, blue) crossings, from Monte Carlo (MC) simulations for the tether geometry and force displayed in **Figure 7**. The positive crossing angle probability distribution was obtained from the MC simulations, whereas the negative crossing angle distribution was derived from the relationship $P_R(\alpha) = P_L(180°-\alpha)$. For Brownian dynamics (BD) simulations, the negative crossing angle distributions, like the positive, were measured directly. The imposed crossing angle ($\alpha_L$: black-dotted line) is the average angle for the positive crossing angle distribution, in this case ≈ 79°. (**B**) Ratio of negative to positive probability from **A** plotted on a semilogarithmic axis. Using the relationship $\tau_L/\tau_R = P_R(\alpha_0)/P_L(\alpha_0)$, the preferred angle ($\alpha_0$) can be obtained, as illustrated by the black arrow. For this tether geometry and force, $\tau_L/\tau_R = 0.19 \pm 0.04$ (obtained from the analysis in **Figure 7C**), which gives $\alpha_0$ of 87.9° ± 4.4° when using MC simulations, and 83.5° ± 4.2° when using BD simulations. The error associated with $\alpha_0$ in **C** and **D** is the combination of the statistical and systematic error, with the latter being the main contributor. (**C**) Preferred crossing angles ($\alpha_0$) from fourteen different DNA tether geometries, plotted against the average positive crossing angles ($\alpha_L$), as measured by MC simulations (filled circles, left-hand plot) and BD simulations (open circles, right-hand plot). (**D**) Preferred crossing angles ($\alpha_0$) from 14 different DNA

*Figure 8 continued on next page*

*Figure 8 continued*

tether geometries, as measured by MC simulations (filled circles, left-hand plot) and BD simulations (open circles, right-hand plot), plotted against the applied force on the DNA tether. The combined average preferred crossing angle for topo VI, determined from the MC simulations, was 87.8° ± 0.4° (± SEM), and from BD simulations, was 87.4° ± 0.4° (± SEM), both values represented by the dotted line and error shading in plots C and D (N = 14).

The online version of this article includes the following source data and figure supplement(s) for figure 8:

**Source data 1.** Source data is in the file *Figure 8*.

**Figure supplement 1.** Estimate of the preferred crossing angle for topo VI from plasmid relaxation ATPase measurements.

**Figure supplement 1—source data 1.** Source data is in the file *Figure 8—figure supplement 1*.

**Figure supplement 2.** Comparisons of crossing angle probability distributions among single-crossings that are relaxed and not relaxed by topo VI.

**Figure supplement 2—source data 1.** Source data is in the file *Figure 8—figure supplement 2*.

---

decatenase. Using MC simulations, it has been shown that angle distributions within catenanes are distributed around 90°, whereas positive supercoils are more acute (*Stone et al., 2003*). A strong crossing angle preference of 87–88° would indeed predispose topo VI to decatenation, while disfavouring supercoil relaxation (*Figure 9*).

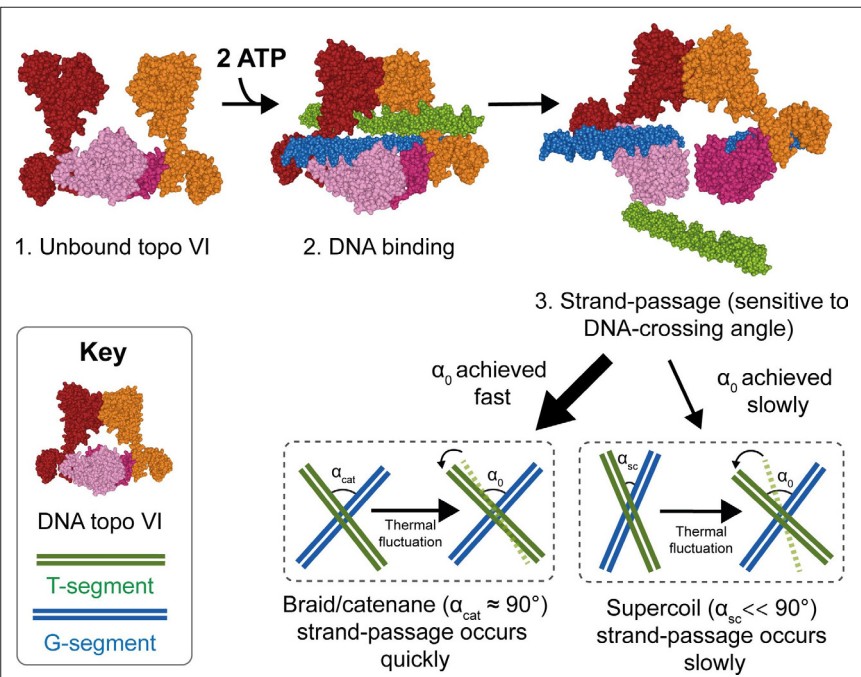

**Figure 9.** Model for chirality-dependent topo VI activity. Unbound topo VI (1) binds a G- and T-segment in the presence of ATP (2), leading to G-segment cleavage and T-segment strand-passage. The rate at which strand passage occurs is sensitive to the DNA-crossing angle. The preferential topo VI DNA-crossing angle ($\alpha_0$) was shown to be ~87.5°, which occurs more frequently in DNA braids and catenanes than in supercoiled DNA. DNA-crossing angles in supercoils ($\alpha_{sc}$) are further from 90° and $\alpha_0$ than the DNA-crossing angles in braids and catenanes ($\alpha_{cat}$), so larger thermal fluctuations are required for supercoils to achieve the preferred topo VI crossing angle, and therefore they are relaxed less efficiently.

## Discussion

### *M. mazei* topo VI is a chirally selective DNA crossing sensor, with preferential decatenase activity

Using the magnetic tweezers single-molecule approach, we have not only demonstrated that *M. mazei* topo VI is a highly distributive and extremely slow supercoil relaxase, confirming findings from *Wendorff and Berger, 2018* (*Wendorff and Berger, 2018*), but have also shown that topo VI activity increases as much as ~10 fold on braided DNA, (*Figures 1, 2 and 4*), approaching rates determined for type IIA topos on their respective optimal substrates (*Agarwal and Duderstadt, 2020*; *Basu et al., 2012*; *Stone et al., 2003*). Along with the observation that topo VI has an extremely strict preference for DNA crossing geometries close to 90°, which appear more frequently in catenanes than they do in supercoils (*Stone et al., 2003*), these data strongly indicate that *M. mazei* topo VI is a preferential decatenase, which simultaneously disfavours supercoil relaxation. This leads to providing a potential explanation for the presence of topo VI in higher eukaryotes, during situations in which the genome is rapidly replicated, such as endoreduplication in plants, explored in depth below.

The preference for positive writhe, explained by the DNA-crossing angle preference of 87.4–87.8°, was an unforeseen result. While significant, with both supercoiled and braided DNA substrates, the 2–3-fold increase in rate on positive writhe was not particularly large, unlike the difference seen for topo IV, which relaxed positive writhe ~20 fold faster than negative (*Neuman et al., 2009*; *Stone et al., 2003*). Whether this difference for topo VI has in vivo implications isn't clear. However, as *M. mazei* encodes a DNA gyrase, which maintains the genome in a negatively supercoiled state, the slight preference for positive DNA crossings may prevent topo VI from interfering with the activity of DNA gyrase. It may also indicate, as supercoils ahead of replication forks are positive, that topo VI can help support the relief of replication- and transcription-induced torsional strain in combination with other topos.

Consideration of the step in the topo VI strand-passage reaction at which chiral selection occurs revealed several possibilities. It could occur during initial DNA binding, T-segment capture, N-gate closure, G-segment cleavage or strand passage. Binding experiments in the absence of ATP indicate that negative DNA crossings are bound more tightly than positive (*Figure 3C and D*), despite both single-molecule and ensemble measurements revealing a two- to threefold rate enhancement in the presence of positive writhe (*Figures 1C–4C*). This suggests that chiral selection does not occur at the initial DNA binding step. One interpretation of the binding data is that topo VI could undergo more G-segment-only binding on negative DNA-crossings, which does not permit activity, however as binding is poor in the presence of relaxed and linearised DNA (*Figure 3C*), it is unlikely that the enhanced binding of negatively supercoiled DNA results from G-segment-only binding. In addition, further confirming observations by *Wendorff and Berger, 2018*, data reported here suggest that topo VI is a DNA-crossing sensor (*Wendorff and Berger, 2018*), e.g. binding the G- and T-segments simultaneously, rather than binding a G-segment first before capturing a T-segment. This model of topo VI is supported by data showing that strand-passage occurs more quickly in plectonemes with more DNA crossings (*Figure 2*), the rate of ATP hydrolysis is tightly coupled to the presence of DNA writhe (*Figure 3A–B*), and DNA binding increases with increasing supercoil density (*Figure 3C–D*). An alternative possibility is that negatively supercoiled DNA binds to topo VI as a product complex, that is with the T segment below the DNA gate, bound in the 'exit cavity' of the enzyme (topo VI does not have an exit gate like type IIA topos). This raises the possibility of strand passage in the opposite direction, that is bottom up, as has been proposed in ATP-independent relaxation of negatively supercoiled DNA by gyrase (*Williams et al., 2001*). However, there is currently no direct evidence to support the ability of topo VI to do this.

It is known from work done by Wendorff and Berger that domains within the Top6B subunit are critical to recognising the T-segment and coupling this to G-segment scission (*Wendorff and Berger, 2018*). It is possible, therefore, that this may not only involve the presence of a T-segment, but more specifically its juxtaposition to the G-segment, which then permits Top6B recognition and N-gate closure. However, in addition to binding enhancement in the presence of negative supercoils, DNA cleavage by topo VI was also enhanced on negative supercoils (*Figure 3C–F*). This suggests that chiral selection does not occur at the N-gate closure or G-segment cleavage steps. This leaves DNA-gate opening and strand passage as the most likely candidates for the chiral discrimination step. There is precedence for this interpretation based on the structure of the type IIA topo, topo IIβ (*Chen et al.,*

*2018*). The structure of the open DNA-gate of topo IIβ, with a fully cleaved and separated G-segment, revealed that the T-segment would likely need to transfer through the break in the left-handed orientation and was suggested to provide further rationale for the type IIA preference for positive-DNA crossings. This may also be true for topo VI, with the likelihood of T-segment transfer being higher for positive DNA-crossings than negative, however in the absence of a structure bound to DNA, this remains speculative.

Another unforeseen outcome of this work was the switch from highly distributive topo VI activity in the presence of supercoils, to mild processivity on braids. There are two main differences between the magnetic tweezers supercoiling and braiding substrates: (1) braids do not accumulate twist upon magnet rotation, while supercoils do, and (2) the DNA crossing angle geometries of braids are closer to 90° than those within the plectoneme (*Figure 6*). So it is likely that one factor, or a combination of both, promotes the processive unlinking of braided DNA substrates by topo VI. The latter is the more likely candidate, with the DNA-crossing geometries found in braids potentially promoting more efficient strand-capture and passage due to the larger imposed crossing angles (*Figure 6*). After the initial strand passage there is a kinetic competition between topo VI releasing the G-segment versus capturing a T-segment and catalysing another strand passage cycle. The larger crossing angles in braids facilitate T-segment capture, resulting in an increase in the number of strand passages catalysed before topo VI releases the G-segment. At low force, which was utilised in the supercoil relaxation measurements (0.4 pN), the increase in twist is minimal as the plectoneme forms after 3–4 magnet rotations (*Figure 1A*). Therefore, it seems unlikely that a minor change in twist would be responsible for the significantly decreased activity on supercoils in comparison to braids. In addition, supercoil relaxation rate initially increased as force, and twist, increased (*Figure 6—figure supplement 1*). The relaxation rate subsequently decreased as the force increased, indicating that twist may become important at higher forces, but this appears to be a secondary factor in determining strand passage rate. Moreover, increasing the force on the DNA braid reduced the strand-passage rate, largely due to increasing dwell times between catalytic events, but also decreasing both the processive rate and the number of strand passages per processive burst (*Figure 6—figure supplement 1A-D*). As twist does not change in the DNA braid, this leaves the DNA geometry as the most likely factor influencing catalysis. As force increases, the average braid angles deviate further from 90° (*Figure 6*), and distributions likely narrow, meaning the probability of achieving the topo VI preferred crossing angle will also decrease. DNA binding and ATPases assays performed here and in *Wendorff and Berger, 2018*, suggest that topo VI preferentially binds and acts on DNA crossings and that G-segment only binding is unfavourable (*Wendorff and Berger, 2018*). However, for processivity to occur, topo VI must remain bound to the initial G-segment once the T-segment has passed. This implies that during the topo VI strand-passage cycle there is competition between the capture and passage of a T-segment at the correct geometry and the rate of dissociation from the DNA, with the former potentially accelerated in the case of DNA braids.

Varying force on the supercoiled substrate suggested that DNA twist could play a secondary role in catalysis. The supercoiling rate initially increases with increasing force before decreasing sharply above a critical force, whereas the twist increases monotonically but sub-linearly with increasing force (*Figure 6—figure supplement 1E-F*; *Neukirch and Marko, 2011*). The initial increase in rate can be explained by the average supercoil angles becoming closer to 90° as force increases (*Figure 6*). The significant decrease in rate that occurs at high force, however, could be due to changes in twist preventing DNA binding or cleavage. However, this could also be due to inhibition of a step that is directly sensitive to force, such as G-segment binding or cleavage, which induce bending of the DNA, or gate closure, particularly since the twist increases sub-linearly with increasing force. Currently, little is known about the dynamics of gate opening for type II topos, but with its relatively simple structure, topo VI could make a good candidate for such studies, like recent experiments with *E. coli* topo I and III that directly observed gate opening dynamics (*Mills et al., 2018*).

## Understanding the activity of topo VI from the archaeal perspective

Archaea constitute the third domain of the tree of life, alongside bacteria and eukarya. Archaea were distinguished from bacteria, previously having been grouped together, based on their unique ribosomal proteins and RNA (*Forterre et al., 2002*). Archaeal DNA metabolism is far less studied than that of the bacterial and eukaryotic domains. However, it is known that archaeal DNA transcription

and replication share features with both bacteria and eukarya (*Ausiannikava and Allers, 2017*; *Barry and Bell, 2006*). For instance, like bacteria, archaea have sequence-specific origins of replication and encode transcriptional regulators that resemble those found in bacteria (*Ausiannikava and Allers, 2017*; *Bell and Jackson, 2001*). But, like the eukarya, archaea utilise a basal transcriptional complex that resembles the eukaryotic RNA polymerase II, can have multiple origins of replication, and many species compact their genomes through the use of histone-like proteins (*Barry and Bell, 2006*; *Bell and Jackson, 2001*). This complexity along with diversity within the archaeal domain itself can make understanding the roles of archaeal topos in vivo more challenging. This is compounded further by the distribution of topos among archaea. Almost all archaea contain topo VI, aside from the order Thermoplasmatales, which instead encodes a DNA gyrase, whereas some, including the Methanosarcina order, encode both gyrase and topo VI (*Forterre and Gadelle, 2009*). In addition to type II topos, all archaea, aside from the Thuamarchaea, encode one or more copies of a type IA topo, which, in terms of sequence similarity, resembles bacterial and eukaryotic topo III, as opposed to bacterial topo IA (*Forterre et al., 2007*). This has led to the proposal that topo VI must be involved in the removal of positive supercoils during transcription and replication, particularly in archaea lacking a gyrase, as topo III is a preferential decatenase (*Lee et al., 2019*). However, the work reported here solidifies the observation by *Wendorff and Berger, 2018* (*Wendorff and Berger, 2018*), that *M. mazei* topo VI is an extremely inefficient DNA relaxase, and provides evidence of its preferential decatenation activity. If this behaviour is true of all archaeal topo VI enzymes, then those organisms that lack DNA gyrase, would struggle to relieve torsional stress ahead of replication forks at a pace required by the cell, if they also do not have an efficient type I relaxase capable of relaxing positive supercoils, like a type IB.

The current literature confounds the issue as the first archaeal type IA topo explored in vitro, from the hyperthermophile *Desulfurococcus amylolyticus*, also termed topo III, was claimed to exhibit robust relaxation activity of both positive and negative writhe at 95 °C. However, it has also been established in *Sulfolobus solfataricus*, that of the three type IA topos it encodes, two are reverse gyrases (*Bizard et al., 2011*) (another type IA topo important in positive supercoiling of hyperthermophillic genomes) (*Nadal, 2007*), and the third is topo III, exhibiting preferential decatenation activity (*Bizard et al., 2018*). It may be that in archaea that lack DNA gyrase and are apparently devoid of a type IA relaxase, that topo VI has evolved more efficient DNA relaxation activity, similar to how *Mycobacterium tuberculosis* DNA gyrase has evolved to efficiently both supercoil and decatenate DNA in the absence of topo IV (*Manjunatha et al., 2002*). *M. mazei* encodes both topo VI and DNA gyrase, as well as two uncharacterised type IA topos, meaning that topo VI may not be required during positive supercoil relaxation and hence has evolved to preferentially decatenate. It is also important, as was done in *Wendorff and Berger, 2018*, to consider the doubling time of the archaea in question, as the slow relaxation by topo VI may be adequate to support the growth of that particular organism, which did not seem to be the case for *M. mazei* (*Wendorff and Berger, 2018*). The involvement of unknown accessory factors that may enhance the rate or processivity of topo VI has been postulated (*Wendorff and Berger, 2018*). In addition, topo VI's main in vivo role could be precatenane removal, which may occur closer to the termination of replication, or even during elongation or at cell division. For instance, if the replication fork can swivel in vivo, which is known to occur in bacteria and eukaryotes (*Cebrián et al., 2015*; *Schalbetter et al., 2015*), positive supercoiling generated ahead of polymerase may be redistributed to form precatenanes behind the fork in archaea, which are likely a preferred topo VI substrate. Therefore, in cells seemingly devoid of an efficient relaxase, topo VI may be able to efficiently support fork progression through unlinking of precatenated DNA, rather than relaxation of supercoils. This may also provide a functional explanation for why topo VI exhibits partially processive behaviour, as this would have utility in the rapid removal of precatenanes. However, as little is currently known about the specifics of how DNA topology changes during archaeal metabolism, this remains speculative. A better understanding of these varied topological states occurring in vivo will also permits more precise delineation of the relative decatenation versus relaxation activities of topo VI. Although topo VI preferentially unlinks rather than relaxes singly-catenated negatively supercoiled substrates (*Figure 5*), in line with its preference for a crossing angle near 90°, the crossing angle distribution in catenated DNA molecules depends on both the sign and extent of supercoiling (*Vologodskii and Cozzarelli, 1996*). Combining the preferred crossing angle obtained here with simulations of the topological states of interlinked DNA in vivo will provide a fuller picture of the relative unlinking versus relaxation activities of topo VI.

## Implications for the physiological role of plant topo VI homologues

Genomic analysis of *Arabidopsis thaliana* revealed that the Topo6A (AtSPO11-3) and Topo6B (AtTOP6B) genes were encoded, and homozygous null mutations in either, displayed identical phenotypes with severely growth stunted plants that failed to thrive (*Hartung et al., 2002*; *Hartung and Puchta, 2000*; *Hartung and Puchta, 2001*). Furthermore, double mutants displayed a phenotype identical to that of either single mutant, demonstrating the proteins likely function in the same process, or even the same protein complex. A process called endoreduplication, in which the genome is replicated multiple times in the absence of cellular division, is critical for plant growth to enlarge hypocotyl and leaf cells (*Sugimoto-Shirasu and Roberts, 2003*), was shown to be deficient in the AtSPO11-3/AtTOP6B mutants and explained the dwarf phenotype (*Robert et al., 2016*; *Vrielynck et al., 2016*). However, *A. thaliana* possesses four type I topos, along with topo II and DNA gyrase, both of which are type IIA topos capable of the same reactions as topo VI, namely DNA decatenation, unknotting, and relaxation (*Corbett and Berger, 2003*). Therefore, the exclusive requirement of topo VI by *A. thaliana* during endoreduplication remains uncertain. Our results provide a possible explanation since topo VI has an intrinsic preference for decatenation, arising from a strict DNA crossing angle preference. Topo VI efficiently unlinks catenanes and braids but exhibits much slower relaxation of supercoils (*Figures 1–4*). During the endocycle, the genome is replicated to variable degrees, however in wild-type *A. thaliana* trichomes this can be as high as 32 C (*Sugimoto-Shirasu and Roberts, 2003*), with mutants in topo VI unable to replicate the genome past 8 C (*Hartung et al., 2002*; *Sugimoto-Shirasu et al., 2002*). With increasing DNA replication, comes elevated levels of transcription, and therefore extensive pressure on the protein machinery involved in both of these processes. Replication- and transcription-dependent strand separation could result in significant levels of DNA supercoiling, which is efficiently dealt with by the type I and type IIA topos. It is possible, therefore, as these proteins are very efficient at relaxation, that even though they are known to efficiently decatenate in vitro, they become subsumed by the necessity to relieve torsional stress generated ahead of replication and transcription forks. With this reaction so heavily disfavoured by topo VI, it could decatenate and unknot the genome unhindered, allowing the cell to continue through the endocycle. The research described here, clearly does not rule out the possibility of protein-protein recruitment, temporal regulation, or other activity-modulating factors which lead to the indispensability of topo VI in endoreduplication, but does provide a rationale that is independent of the requirement of these yet unknown components.

# Materials and methods

## Key resources table

| Reagent type (species) or resource | Designation | Source or reference | Identifiers | Additional information |
|---|---|---|---|---|
| Gene (*Methanosarcina mazei*) | Top6A | Gift from James Berger, Johns Hopkins University | NCBI Gene ID: 1480760 | |
| Gene (*Methanosarcina mazei*) | Top6B | Gift from James Berger, Johns Hopkins University | NCBI Gene ID: 1480759 | |
| Strain, strain background (*E. coli*) | Rosetta 2 (pLysS) | Novagen | | |
| Genetic reagent (*E. coli*) | pBR322* | Inspiralis | | |
| Genetic reagent (*E. coli*) | pET28a | EMD Millipore | CAT#: 69,865 | |
| Genetic reagent (*E. coli*) | pBlueScript II KS(+) | Agilent | CAT#: 212,207 | |
| Recombinant DNA reagent | *Top6AB* dual expression vector | PMID:17603498 | | *Corbett et al., 2007* |

*Continued on next page*

*Continued*

| Reagent type (species) or resource | Designation | Source or reference | Identifiers | Additional information |
|---|---|---|---|---|
| Antibody | Anti-digoxigenin (Sheep Polyclonal) | Roche | Roche Cat# 11333089001, RRID:AB_514496 | Reconstituted in 1 x Phosphate buffered saline (0.6 µg) |
| Commercial assay or kit | PCR DNA purification kit | Qiagen | Qiagen Cat. #: 28,104 | |
| Other | Streptavidin coated magnetic beads (ø: 1 and 2.8 µm) | Invitrogen | Invitrogen Cat. #: 65,602 and 65,305 | |
| Chemical compound, drug | Phusion high-fidelity DNA polymerase | New England Biolabs | NEB Cat. #: M0530 | |
| Chemical compound, drug | T4 DNA ligase | Promega | Promega Cat. #: M1801 | |
| Chemical compound, drug | BsaI-HF | New England Biolabs | NEB Cat. #: R3535 | |
| Chemical compound, drug | Biotin-16-dUTP | Roche | Sigma Cat#: 11093070910 | |
| Chemical compound, drug | Digoxigenin-11-dUTP | Roche | Sigma Cat#: 11558706910 | |
| Sequence-based reagent | 5 kb DNA supercoil primer1 | Eurofin Genomics *Seol and Neuman, 2011a* | | 5'- GCT GGG TCT CGG TTG TTC CCT TTA GTG AGG GTT AAT TG |
| Sequence-based reagent | 5 kb DNA supercoil primer2 | Eurofin Genomics *Seol and Neuman, 2011a* | | 5'- GCT GGG TCT CGT GGT TTC CCT TTA GTG AGG GTT AAT TG |
| Sequence-based reagent | 3 kb DNA braid primer1 | Eurofin Genomics | | 5'(2 x)biotin-GCTGGGTCTCGGTT GGAACTGCGACT GGATAGG |
| Sequence-based reagent | 3 kb DNA braid primer 2 | Eurofin Genomics | | 5' (3 x) digoxigenin-GCTGGGTCTCGGTT GGATTACGCCA GTTGTACG |
| Sequence-based reagent | 5 kb DNA braid primer1 | Eurofin Genomics | | 5'(2 x)biotin-CTTCCGCTTCCTC GCTCACTGACTC |
| Sequence-based reagent | 5 kb DNA braid primer 2 | Eurofin Genomics | | 5' (3 x) digoxigenin-CTGTTCATCCGC GTCCAGCTCGTTG |
| Sequence-based reagent | Bio/Dig labelled Primer1 | Eurofin Genomics *Seol and Neuman, 2011a* | | 5'-GGA CCT GCT TTC GTT GTG GCG TAA TCA TGG TCA TAG |
| Sequence-based reagent | Bio/Dig labelled Primer2 | Eurofin Genomics *Seol and Neuman, 2011a* | | 5'- GGG TCT CGT GGT TTA TAG TCC TGT CGG GTT TC |
| Software, algorithm | LabVIEW, Instrument control software | National Instruments | NI Cat. #: 776678–35 | |
| Software, algorithm | Igor Pro 7, Data analysis | WaveMetrics | PMID:28069956 | |
| Software, algorithm | ImageJ, Data analysis | National Institutes of Health | | |
| Chemical compound, drug | Adenosine triphosphate (ATP) | MilliporeSigma | A2383 | |

*Continued on next page*

*Continued*

| Reagent type (species) or resource | Designation | Source or reference | Identifiers | Additional information |
|---|---|---|---|---|
| Chemical compound, drug | Nicotinamide adenine dinucleotide (NADH) | MilliporeSigma | 10107735001 | |
| Chemical compound, drug | Pyruvate Kinase/Lactic Dehydrogenase (PK/LDH) | MilliporeSigma | P0294 | |
| Chemical compound, drug | Phosphoenol-pyruvate (PEP) | MilliporeSigma | 10108294001 | |
| Other | Plate reader | BMG LabTech | CLARIOstar Plus | Used for the PK/LDH-coupled ATPase assay. |
| Software, algorithm | Microsoft Excel | RRID:SCR_016137 | | Used for data analysis for the PK/LDH-coupled ATPase assay. |
| Software, algorithm | LAMMPS | https://www.lammps.org/ | | Used for Molecular Dynamics Simulations |
| Software, algorithm | MATLAB | MathWorks | | Used for analyses of Molecular Dynamics Simulations |

## Protein expression and purification

Both subunits of *M. mazei* topo VI (*top6A* and *top6B*) were expressed from a polycistronic dual expression vector, kindly provided by James Berger (*Corbett et al., 2007*), transformed into Rosetta 2(DE3)pLysS Singles Competent Cells (Novagen). Cells were grown for 24 hr at 37 °C in autoinduction growth media (AIM) with kanamycin (50 mg/ml) and chloramphenicol (35 mg/ml). The culture was then centrifuged using the RC6+ centrifuge (Sorvall) in a FS9 rotor for 8 min, at 8000 rpm, at 4 °C. Supernatant was discarded and the pellet resuspended in Buffer A (20 mM HEPES pH 7.5, 10% (v/v) glycerol, 800 mM NaCl, 20 mM Imidazole, 2 mM β-mercaptoethanol and cOmplete EDTA-free protease inhibitors (Roche)).

Cells were lysed under high pressure using an Avestin high pressure homogeniser. Samples were then spun at 18,500 rpm for 1 hr at 4 °C with the RC 6+ centrifuge (Sorvall) and SS34 rotor. The lysate was then passed over a HisTrap FF Ni$^{2+}$ column (5 ml/min, GE Life Sciences) and washed with Buffer B (20 mM HEPES pH 7.5, 10% (v/v) glycerol, 150 mM NaCl, 20 mM Imidazole, 2 mM β-mercaptoethanol, and cOmplete EDTA-free protease inhibitors (Roche)). The protein was stepped off in Buffer B1 (20 mM HEPES pH 7.5, 10% (v/v) glycerol, 150 mM NaCl, 500 mM imidazole, 2 mM β-mercaptoethanol, and cOmplete EDTA-free protease inhibitors (Roche)) and then loaded on to a HiTrap SP Sepharose HP column (5 ml/min, GE Life Sciences) followed in tandem by a HiTrap Q Sepharose HP column (5 ml/min, GE Life Sciences). The SP Sepharose column was removed before the protein was stepped off from the Q sepharose with Buffer B2 (20 mM HEPES pH 7.5, 10% (v/v) glycerol, 800 mM NaCl, 20 mM Imidazole, 2 mM β-mercaptoethanol). Protein concentration was assessed using solution absorbance at 280 nm, determined using a Nanodrop. The his-tag was cleaved using tobacco etch virus protease (TEV) at a 1:50 concentration ratio (TEV:topo VI) overnight at 4 °C. The cleaved protein was then passed over a HisTrap FF Ni$^{2+}$ column (5 ml/min, GE Life Sciences) and washed with Buffer B and the cleaved topo VI collected in the flow through. The un-cleaved topo VI, his-tag and his-tagged TEV were then stepped off in Buffer B1. Cleaved topo VI fractions were pooled and concentrated using Amicon Ultra 15 mL centrifugal filter units (30 kDa cut off, MilliporeSigma) before being passed down a Superose 6 10/300 (GE Life Sciences) column in Buffer C (20 mM HEPES pH 7.5, 10% (v/v) glycerol, 300 mM NaCl, 2 mM β-mercaptoethanol and cOmplete EDTA-free protease inhibitors (Roche)), subsequently concentrated and topo VI concentration determined using a Nanodrop. All proteins stored at –80 °C. The above protocol was adapted from *Corbett et al., 2007* (*Corbett et al., 2007*).

## Single molecule magnetic tweezers

The magnetic tweezers instrumentation used here has been detailed extensively elsewhere (*Charvin et al., 2003*; *Dittmore et al., 2017*; *Seol and Neuman, 2011a*; *Seol and Neuman, 2011b*). Here,

both DNA braids and supercoils have been utilised. DNA duplexes (5 kb for supercoiling and 3 kb for braiding) are tethered at one extreme end to a glass slide by the interaction between digoxigenin-labels on the DNA and the anti-digoxigenin-coated glass slide. A streptavidin-coated magnetic bead (1 µm diameter, Dynabeads MyOne Streptavidin T1; 35601, Invitrogen) is bound to the opposite biotinylated end of the DNA. Rotation of the fixed magnets above the sample cell caused rotation of the magnetic bead, which changed the linking number of the bound DNA molecule. For super-coiled DNA experiments, both DNA ends were bound to both the surface of the sample cell and the magnetic bead via multiple attachment points so that they are torsionally constrained and accumulate twist and writhe upon magnet rotation. For braided DNA experiments, two bound DNA duplexes are necessary; however, as they have a single digoxigenin labelled nucleotide, which is rotationally uncon-strained, the rotation of the magnets cause the duplexes to wrap around one and other, increasing writhe, while twist is dissipated by the free rotation of the DNA.

Magnetic tweezers measurements were conducted with 0.05–2 nM of *M. mazei* topo VI in 20 mM Bis-Tris propane (pH 7), 100 mM potassium glutamate, 10 mM MgCl$_2$, 1 mM DTT, 0.01% tween-20, 30 µM bovine serum albumin (BSA) and 1 mM ATP at 21 °C, using 0.2–4 pN of force. The magnetic tweezers software was run using Lab view, with a data sample rate of 200 Hz, and analysis of the magnetic tweezers data was done using a *t*-test based method, described in *Seol et al., 2016*, with the routine written for use in IGOR pro 7 (Wavemetrics). For further detail on calibration and analysis of single-crossing data see *Neuman et al., 2009*.

## DNA simulations

Monte Carlo (MC) DNA simulations were conducted in order to obtain DNA-crossing angle distribu-tions for use in determination of the topo VI preferred crossing angle as described in *Neuman et al., 2009* (*Charvin et al., 2005*; *Neuman et al., 2009*).

We confirmed the MC simulation approach by Brownian dynamics simulations of the same DNA braids with identical parameters (DNA length, spacing between DNA molecules, and force). We performed Brownian dynamics (BD) simulations of two 3000 bp long DNA molecules using an estab-lished coarse-grained model (*Brackley et al., 2013*; *Jun and Mulder, 2006*; *Kim et al., 2015*; *Le Treut et al., 2016*; *Pereira et al., 2017*). We provide a brief description of the simulation model below. For a more detailed description of the simulation model please refer to *Pereira et al., 2017*.

Briefly, DNA is simulated as a linear polymer consisting of charged monomers. Each monomer represents 7.4 bp and carries a charge of $-2.96e$. Below we provide a description of the simulation model in reduced Lennard-Jones (LJ) units. The interaction between monomers is defined by four contributions.

$$U_{total} = U_{FENE(r_{i,i+1})} + U_{BEND(\theta_{i-1,i,i+1})} + U_{WCA(r_{ij})} + U_{DH(r_{ij})}$$

Here,

$$U_{FENE(r_{i,\ i+1})} = -0.5 K_{FENE} R_0^2 ln \left[ 1 - \left( \frac{r_{i,i+1}}{R_0} \right)^2 \right]$$

Describes the finitely extensible non-linear elastic (FENE) bond interaction between two consecu-tive monomers ($i$ and $i + 1$) along the polymer chain separated by a distance $r_{i,\ i+1}$. $K_{FENE} = 30 k_b T/\sigma^2$ is the spring constant and $R_0 = 1.6\sigma$ defines the maximum bond length. Here, $\sigma$ is the reduced unit of length and approximately equal to 2.5 nm in real units. The bending rigidity of the DNA is defined by

$$U_{BEND\ (\theta_{i-1i,i+1})} = K_{BEND} \left( 1 + cos \left( \theta_{i-1,i,i+1} \right) \right)$$

Here, $\theta_{i-1,i,i+1}$ is the angle between three consecutive monomers along the polymer chain and $K_{BEND} = 20 k_B T$ is the bending rigidity.

The excluded volume interactions between non-bonded monomers ($i$ and $j$) separated by a distance of $r_{ij}$ is defined by Weeks-Chandler-Andersen (WCA) potential:

$$U_{WCA}(r_{ij}) = \begin{cases} 4k_BT\left[\left(\frac{\sigma}{r_{ij}}\right)^{12} - \left(\frac{\sigma}{r_{ij}}\right)^{6}\right] + k_BT, & r_{ij} < 2^{\frac{1}{6}}\sigma \\ 0, & r_{ij} \geq 2^{\frac{1}{6}}\sigma \end{cases}$$

The electrostatic interactions between monomers are calculated using the Debye-Huckle potential ($U_{DH(r_{ij})}$),

$$U_{DH(r_{ij})} = \begin{cases} k_BTl_b\frac{q_iq_j}{\epsilon r_{ij}}e^{-\frac{r_{ij}}{l_b}}, & r_{ij} < 6\sigma \\ 0, & r_{ij} \geq 6\sigma \end{cases}$$

where, $l_b$ is the Bjerrum length, $\epsilon$ is the dielectric constant and $q_i$ is the charge on particle . We set $l_b^{-1} = 3.66\sigma$ and $\epsilon = 1.6$ to match the experimental salt concentration.

The separation between the two DNA molecules, the force applied on the DNA and the number of turns applied were set to match experimental conditions for each experiment. BD simulations were carried out in a constant number of particles, volume, and temperature (NVT) ensemble. Constant temperature is maintained using a Langevin thermostat. Simulations were carried out using a timestep of $0.01\tau$, here $\tau$ is the reduced LJ unit of time. The simulations were performed for $4 * 10^7\tau$ and data was collected every $250\tau$. All the BD simulations were performed using Large-scale Atomic/Molecular Massively Parallel Simulator (LAMMPS) (*Plimpton, 1995*).

The crossing angles between the two DNA molecules in the molecular dynamics and Monte Carlo simulations were calculated in the same manner. Following established methods (*Charvin et al., 2005*; *Neuman et al., 2009*; *Vologodskii and Cozzarelli, 1996*; *Vologodskii and Cozzarelli, 1994*), a juxta-position event was considered to take place when the separation between the two DNA chains was less than a threshold distance (10 nm). The angle formed by the two segments of closest approach was determined by considering the clockwise rotation that would align the bottom strand with the top strand.

## Agarose gel-based DNA relaxation and decatenation
Topo VI was diluted using Dilution Buffer (20 mM HEPES (pH 7.5), 10% (v/v) glycerol, 100 mM potassium glutamate and 2 mM β-mercaptoethanol) for use at final concentrations of 0.1–40 nM, for agarose gel-based assays. Reactions were run in a 30 µL reaction volume with Relaxation Buffer (20 mM Bis-Tris propane (pH 7), 100 mM potassium glutamate, 10 mM MgCl₂, 1 mM DTT and 1 mM ATP) and 2.5 nM negatively supercoiled pBR322* (Inspiralis) or singly-linked catenanes (Inspiralis) (see (*Waraich et al., 2020*) for details on catenane substrate preparation), for use in relaxation or decatenation assays, respectively. Reactions were incubated at 22 or 37 °C for 30 min (unless otherwise stated, e.g. time course) then stopped by the addition of 20 µL of 2 x Loading Buffer (100 mM Tris·HCl pH 8.0, 40% sucrose, 100 mM EDTA, 0.5 mg/ml Bromophenol Blue) and 30 µL chloroform:isoamyl alcohol (24:1 v/v). The reaction mixture was vortexed and centrifuged for 3 min in an Eppendorf 5,425 centrifuge at 21,000 rcf, before being loaded onto a native 1% (w/v) TAE agarose gel (40 mM Tris·HCl pH 8.5, 20 mM glacial acetic acid and 1 mM EDTA) and run for 15 hr at 2 Vcm⁻¹. The gels were stained for 1 hr using 0.5 µg/mL ethidium bromide in TAE, and destained for 30 min in TAE alone, before being imaged via UV transillumination.

## Agarose gel-based DNA cleavage
The DNA cleavage assay was performed as per the relaxation/decatenation assays with the following modifications. Topo VI (5–80 nM) was incubated with 2.5 nM negatively supercoiled pBR322* (Inspiralis) in a 30 µL reaction volume with Cleavage Buffer (20 mM Bis-Tris propane (pH 7), 100 mM potassium glutamate, 10 mM MgCl₂, 1 mM DTT and 1 mM ADPNP) for 30 min at 37 °C. The reaction was stopped with the addition of 0.2% SDS and 3 units of Proteinase K (New England Biolabs, #P8107S), and incubated for 1 hr at 37 °C. The assay was then treated the same as for relaxation and decatenation assays, with the addition of 2 x Loading Buffer and chloroform:isoamyl alcohol (24:1 v/v), followed by agarose gel electrophoresis.

## ATPase assays

Topo VI ATPase activity was measured using two methods: a PK/LDH-linked assay described previously (*Feng et al., 2021*; *Figure 3—figure supplement 1*) and a radioactive ATP assay (*Figure 3A*). For the radioactive assay, an ATP purification column was prepared using P-2 gel (Biorad). In 10 mL of P-2 wash buffer (10 mM Tris·HCl pH 8, 50 mM NaCl, 1 mM EDTA), 1.67 g of P-2 was added and left to hydrate overnight at 21 °C. The supernatant was removed from the hydrated gel, which was then washed four times with degassed P-2 wash buffer. The gel was then poured into a 4 mL Econo-Column (Biorad) and allowed to settle before P-2 buffer was slowly pumped through the column at 0.2 mL/min. A 100 µL ATP solution was made using 97 µL of 100 mM ATP, 2 µL of 33 mM [γ-$^{32}$]-ATP and 1 µL of a 1% (w/v) bromophenol blue solution dissolved in DMSO. Any residual P-2 wash buffer was carefully removed without disturbing the column matrix and the ATP solution added. Once the ATP solution had fully entered the column, the column was filled to the top with P-2 buffer. The ATP solution was left to migrate down the column, and using a Geiger counter pointed at the base of the column, the elution of the [γ-$^{32}$]-ATP was monitored. Once the [γ-$^{32}$]-ATP was eluting from the column, single drops were collected as separate fractions until the radioactivity signal intensity dropped again. Using TLC plates, 0.5 µL drops were spotted from each [γ-$^{32}$]-ATP fraction 1 cm from the bottom of the plate. The plate was then positioned upright with the [γ-$^{32}$]-ATP spots at the bottom in TLC running buffer (0.5 M lithium chloride in 1 M glacial acetic acid). Once the running buffer had migrated at least two thirds up the TLC plate, the plate was removed and fully dried. The dry TLC plates were then exposed for 1 hr to a phosphor screen, which was then imaged with a Typhoon FLA 7000 plate reader. The fractions shown to contain the most [γ-$^{32}$]-ATP and the least [γ-$^{32}$]-phosphate were pooled and the concentration determined using absorbance at 260 nm (ATP $\varepsilon$ = 15,400 M$^{-1}$cm$^{-1}$).

Using the purified [γ-$^{32}$]-ATP solution, a topo VI relaxation time course was performed. The reaction constituents were as follows: 1 µM topo VI, 430 nM DNA (relaxed, linear or supercoiled pBR322*), 450 µM [γ-$^{32}$]-ATP, 10 mM MgCl$_2$ and 1 x minimal buffer (20 mM bis-Tris propane (pH 7), 100 mM potassium glutamate and 1 mM DTT), in a 55 µL reaction volume total. At each time point (1, 2, 4, 6,8, 10, 15, 20, and 30 minutes) a 5 µL aliquot was taken from the reaction and quenched using 5 µl of 2% SDS and 100 mM EDTA. A 1 µL aliquot of each time point was then spotted onto a TLC plate and treated as above. The intensity of the spots were calculated using ImageJ and the portion of ATP hydrolysed at each time point calculated using the proportion of phosphate released.

## DNA binding

To measure the binding of topo VI to DNA in the absence of nucleotide, a nitrocellulose membrane capture technique was employed. It is described in detail in *Litwin et al., 2015*, with slight modifications. Briefly, in a 500 µL volume, 16 nM topo VI was incubated with 10 nM pBR322* of varying topology (linear, negatively supercoiled, positively supercoiled and relaxed, obtained from Inspiralis) in Binding Buffer (20 mM bis-Tris propane (pH 7), 100 mM potassium glutamate, 10 mM MgCl$_2$ and 1 mM DTT), for 30 minutes at 37 °C. The reaction was then added to a 0.45 µm nitrocellulose Centrex MF filter (Whatman) equilibrated with binding buffer, and centrifuged at 2000 rpm for 5 min. The flow through, containing unbound DNA, was removed and saved, and 500 µL of binding buffer added before a further centrifugation under the above conditions. The wash was removed and 500 µL of 10 mM Tris·HCl (pH 7.5) and 0.2% sodium dodecyl sulphate (SDS) was added to the spin column and centrifuged once more, to elute topo VI-bound DNA. The fractions containing bound and unbound DNA were concentrated and buffer exchanged into 10 mM Tris·HCl (pH 7.5) using 30 kDa Amicon Ultra-0.5 columns (Millipore). The DNA concentration was measured using the nanodrop and the percentage bound of the total was calculated.

To further this analysis, distributions of positive and negative topoisomers were generated using *Archaeoglobus fulgidus* reverse gyrase. In 50 µL, 11 nM negatively supercoiled DNA was incubated with 100 nM reverse gyrase, 1 mM ATP, in 50 mM Tris·HCl (pH 8.0), 10 mM NaCl and 10 mM MgCl$_2$, at 95 °C for 10 or 20 sec (to generate negative and positive topoisomer distributions respectively) and stopped using 10 µL 0.5 M EDTA. To clean up the DNA, 2 µL of 2% SDS and 2 µL of proteinase K (New England Biolabs) were added and the solution was incubated at 37 °C for 1 hr before purification with the Qiagen DNA clean up kit. The topology was confirmed by running 60 ng of the positively and negatively supercoiled DNA on a 1% (w/v) agarose gel containing 1.5 µg/mL chloroquine. It was then used in binding assays as described above, however once the topo VI-bound DNA was cleaned up

and quantified, it was run on a native 1% (w/v) agarose gel for 15 hr at 2 Vcm$^{-1}$. The gels were stained for 1 hr using 0.5 µg/mL ethidium bromide in TAE, and destained for 30 min in TAE, before being imaged via UV transillumination. The intensity of each topoisomer band was determined using ImageJ and the relative $K_d$ values for each topoisomer were calculated as described in *Litwin et al., 2015*.

### Data analysis and figure preparation

Graphs were made using Igor Pro 7 (Wavemetrics) and figures were assembled using Adobe Illustrator.

## Acknowledgements

The authors thank James Berger for the *M. mazei* topo VI expression plasmid, James Taylor for his guidance on the radioactive ATPase assays, Lipeng Feng for his help with the PK/LDH ATPase assay and Lesley Mitchenall for her expertise and training. SJM was supported by the intramural program of the National Heart, Lung, and Blood Institute, National Institutes of Health, and by the Wellcome Trust. This work utilized the computational resources of the NIH HPC Biowulf cluster (http://hpc.nih.gov). Work in KCN's lab was supported by the intramural research program of the National Heart, Lung, and Blood Institute, National Institutes of Health, Department of Human Services. Work in AM's lab was supported by the Biotechnology and Biosciences Research Council (UK) Institute Strategic Programme Grant BB/P012523/1, and the Wellcome Trust (Investigator Award 110072/Z/15/Z).

## Additional information

### Funding

| Funder | Grant reference number | Author |
|---|---|---|
| National Institutes of Health | 1ZIAHL001056 | Shannon J McKie<br>Parth Desai<br>Yeonee Seol<br>Keir C Neuman |
| Wellcome Trust | | Shannon J McKie |
| Biotechnology and Biological Sciences Research Council | BB/P012523/1 | Anthony Maxwell |
| Wellcome Trust | 110072/Z/15/Z | Anthony Maxwell |

The funders had no role in study design, data collection and interpretation, or the decision to submit the work for publication.

### Author contributions

Shannon J McKie, Conceptualization, Data curation, Formal analysis, Investigation, Methodology, Project administration, Resources, Writing – original draft, Writing – review and editing; Parth Rakesh Desai, Investigation, Methodology, Software; Yeonee Seol, Investigation, Methodology, Project administration, Resources, Software, Supervision, Writing – review and editing; Adam MB Allen, Investigation, Writing – review and editing; Anthony Maxwell, Conceptualization, Formal analysis, Funding acquisition, Investigation, Methodology, Project administration, Resources, Supervision, Writing – review and editing; Keir C Neuman, Conceptualization, Formal analysis, Funding acquisition, Investigation, Methodology, Project administration, Resources, Supervision, Validation, Writing – original draft, Writing – review and editing

### Author ORCIDs

Anthony Maxwell ![ORCID] http://orcid.org/0000-0002-5756-6430
Keir C Neuman ![ORCID] http://orcid.org/0000-0002-0863-5671

### Decision letter and Author response

Decision letter https://doi.org/10.7554/eLife.67021.sa1
Author response https://doi.org/10.7554/eLife.67021.sa2

## Additional files

### Supplementary files
• Transparent reporting form

### Data availability
All data are provided in the source files associated with each figure and figure supplement.

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
