## [Editor Report]

The present work is noteworthy for explaining how DNA topoisomerase VI, an archaeal and plant based enzyme with homology to the Spo11 meiotic recombination core complex, senses DNA crossovers to preferentially remove positive supercoils and DNA catenanes. The findings are important for understanding how topoisomerase VI supports DNA replication and chromosome disentanglement.

---

## [Decision Letter]

**Decision letter after peer review:**

Thank you for submitting your article "Topoisomerase VI is a chirally-selective, preferential DNA decatenase" for consideration by *eLife*. Your article has been reviewed by 2 peer reviewers, one of whom is a member of our Board of Reviewing Editors, and the evaluation has been overseen by Cynthia Wolberger as the Senior Editor. The reviewers have opted to remain anonymous.

Essential revisions:

Please address the points raised by the referees in the sections below.

*Reviewer #1:*

In current paper, McKie et al. describe single-molecule magnetic tweezer measurements to show that M. mazei topoisomerase VI relaxes DNA more effectively when acting on positively supercoiled substrates, and that it unlinks DNA catenanes (experimentally mimicked by forming braided DNAs), particularly substrates with positive crossings, even more actively. Following an established approach used to characterize the chiral discrimination of DNA by bacterial topoisomerase IV, the authors integrate the magnetic tweezer data with angular probabilities derived from computational simulations of fluctuating DNA crossings. It is established that the penchant for specific DNA crossing angles by topo VI underlies the enzyme's preference for positive crossings, and that those angles are found more frequently in braids/catenanes than in plectonemes. A combination of magnetic tweezer measurements and biochemical assays are also used to show that while topo VI is a distributive and slow relaxase, it is a more potent and processive decatenase.

McKie et al. also deduce constraints on the mechanism of topo VI, beyond the preferred crossing angle, by interpreting counterintuitive results of DNA binding and cleavage assays. The tweezer data show greater relaxation activity for positive supercoils, yet the strength of DNA binding and amount of DNA cleavage are higher for negative supercoils. The authors rule out G-segment-only binding on negative supercoils (since binding and ATPase activity on relaxed and linear DNA are poor) and instead posit that chiral preference is set at the DNA-gate opening or strand passage steps, as has been proposed for human topo II β.

The paper is well-organized and clearly written, with observations appropriately contextualized in light of previous work showing that topo VI is a DNA crossover sensor, and with respect to other type II topoisomerases and the biology of archaea and plants. While there are some suggested recommendations that can be made for clarifying certain points or shoring up some of the claims, the findings overall appear supported by the data. The work constitutes a meaningful contribution to the understanding of mechanism in topo VI and the results and interpretations are situated thoughtfully amid the existing type II topoisomerase literature.

• The authors have specified that error bars refer to standard error of the mean but haven't stated numbers or natures of replicates.

• It would be helpful to show ATPase activity data for positively supercoiled DNA as well as negative in Figure 4A.

• In the Discussion section, page 23 paragraph 2, it says that the near-90° crossing angle for braids could explain the switch from distributivity on plectonemes to processivity on braids by "promoting more stable DNA binding and efficient strand passage." This is speculative and a bit at odds with the binding on supercoils, and further raises the question as to whether there is a way to show binding strength (or unbound/bound fractions) for braids/catenanes in parallel with the data for negative supercoil binding tighter than positive. Please comment.

• Page 14 paragraph 1 (in regard to Figure 4D) states that the rate and extent of processive bursts are fairly constant over large concentration changes. However, for the rate plot, there is by eye a notable, fairly steady rate dependence (~1.5x increase for positive, ~2x increase for negative), especially in light of how the changes in Figure 6 Figure Supplement 1 C and D are described. It might be better to admit that the rates do increase slightly, but that since the burst sizes do not increase and because the dwell times decrease to some limit, it could still be concluded that the phenomena result from a single, processive enzyme rather than a concentration increase.

• Page 8, end of first paragraph. Why assume that positive crossover selection occurs following DNA cleavage? Why not just posit that the binding of negative crossovers, while favored, is also mildly inhibitory to strand passage? This would seem very analogous to topo IV. Or perhaps there is some confusion as to when preference occurs that could be cleared up in the writing?

*Reviewer #2:*

Diverse type II topoisomerases perform critical roles managing DNA topology in cells; all off these enzymes are capable of carrying out a duplex strand passage reaction, but different enzymes specialize in harnessing this mechanism for distinct functions such as relaxation of positive and/or negative supercoiling, introduction of negative supercoiling, or decatenation. McKie et al. set out to characterize and understand the specialization of topoisomerase VI. They make use of tools including single-molecule assays with exquisitely well-controlled substrates to build on prior structural and biochemical work and help reconcile prior reports of its weak supercoil relaxation activity with its physiological importance.

The central comparison of single-molecule activity on supercoiled vs braided DNA molecules confirms and extends prior characterization of topoisomerase VI as a weak and distributive supercoil relaxase, while simultaneously clearly establishing that braids – an analog of physiological catenanes – are much more robustly relaxed. The application of the elegant single-crossing assay (introduced by Neuman et al. in earlier work), interpreted with the help of MC and BD simulations, leads to the strongly supported finding that topoisomerase VI prefers a crossing angle slightly less than 90 degrees. The authors also obtain a very clear measurement of a modest chiral preference on both supercoiled and braided substrates, and it is satisfying that for this enzyme the optimal crossing angle alone appears sufficient to explain this preference.

It is clear that there are relatively high average rates of braid relaxation. The stronger claim of even faster processive bursting may well also be correct, but is harder to evaluate from the information presented, which does not include a detailed analysis of the limits of the step-finding procedure (from Seol and Neuman) in its specific application here, nor any extensive presentation of raw traces. Given the quite small reported processivities, small extension signals in comparison to supercoil relaxation, and burst rates that are not quite insensitive to concentration, this is a claim where the details of the analysis are important for establishing confidence.

It is striking that the preferred crossing angle is quite similar to topo IV, but the degree of preference seems much more stringent – a conclusion based on the narrow range of geometries where the authors observe activity in single-crossing assays. A stringent preference for crossing angles near 90 degrees is central to the authors' explanations of preferential activity on braids and catenanes. However, the characterization and explanation of the difference in stringency with topo IV seem underexplored quantitatively, since the authors do not discuss e.g. the raw probabilities of sampling the optimal crossing angle in the assayed geometries that did not show any measurable activity, or the limits of detectable rates in their measurements.

Overall, this work advances the field by presenting a compelling case for topo VI as an effective decatenase, helping focus attention on this activity in its physiological roles, while presenting a plausible hypothesis – a strict crossing angle preference for strand passage – uniting observations of activities on diverse substrates.

To address my comments about burst analysis, I suggest:

1. Showing steps of representative traces, together with overlaid step-finding outputs.

2. Providing details of the step-finding procedure and subsequent analysis. E.g, the legend indicates that bursts were defined as two or more cycles; at what stage was that criterion imposed? Did step-finding also find smaller bursts (e.g. 1 cycle) and then those were excluded from subsequent analysis? Were the reported burst sizes corrected to account for the detection threshold?

3. Revisiting the simulation approach that was compelling for validating results in Seol et al. For example, if you simulate a "null hypothesis" of distributive activity with the same *average* braid relaxation rate measured in your assay, and include realistic noise, then apply the analysis pipeline used for experiments, what comes out?

To address my comments about ingle-crossing stringency, I suggest:

1. Report directly on tethers that showed no activity; what was the simulated distribution of crossing angles for those tethers? In particular, how does the probability of sampling the preferred crossing angle compare with tethers for which activity was observed?

2. How much slower must the rates be on these "no activity detected" tethers for relaxation to not be measurable?

3. While not as elegant as the internally controlled left-right comparisons, can raw rates of relaxation be compared between different tethers and related to the simulated probabilities of observing the preferred crossing angle?

Additional questions and suggestions:

Generally the discussion of crossing angles for different substrates (braids and supercoils at difference forces, catenanes and supercoils in the absence of force) would be quantitatively improved by calculating *distributions* of crossing angles for these cases in a unified analysis; I recognize this is a significant simulation task and currently you reference some past work to support general claims. At minimum I would reference, consider, and discuss early work presenting distributions of juxtaposition angles, from Vologodskii and Cozzarelli (1994 and 1996), which includes simulations of weakly supercoiled DNA (1996) where the juxtapositions that occur are still broadly distributed and just starting to be asymmetric.

The biochemical decatenation assay used here and in Ref. 47 is quite striking. What can you say about how "special" topo VI is in terms of the results of that assay? Ref. 47 shows a similar result for topo II α, where catenation occurs at a lower enzyme concentration than supercoil relaxation. How about other topo II enzymes, e.g. yeast topo II, bacterial topo IV…?

[Editors' note: further revisions were suggested prior to acceptance, as described below.]

Thank you for resubmitting your work entitled "Topoisomerase VI is a chirally-selective, preferential DNA decatenase" for further consideration by *eLife*. Your revised article has been reviewed by 2 peer reviewers, one of whom is a member of our Board of Reviewing Editors, and the evaluation has been overseen by Philip Cole as the Senior Editor.

The manuscript has been much improved. Both referees are favorable but request that a couple of issues still be addressed, as outlined below:

*Reviewer #1:*

The authors have satisfactorily addressed all major comments. A couple typos still appear to have snuck through: "WaveMentrics" in Figure 1C caption, and the formatting of Weeks-Chandler potential is fixed but the Debye-Huckle is not.

*Reviewer #2:*

In their revision, McKie et al. have added substantial new analyses, experiments, and clarifications to their manuscript, strengthening their study while responding thoughtfully to many of the issues raised by both reviewers. I am supportive of publication in *eLife*, and my remaining comments are concerned with just a few points on which I feel the authors could be more complete in addressing questions from the previous round of review, without requiring additional experiments or extensive calculations.

Burst analysis for unbraiding analysis:

I previously asked that the authors show representative traces, together with overlaid step-finding outputs. This request was minimally addressed by adding the step-finding overlay to the single trace that was already included in Figure 4B. To give the reader a more representative sense of this data and analysis, I think it would be appropriate to show more traces and overlays, including under varying conditions; this could be done in a supplementary figure. The annotation and description of Figure 4B itself could also be more clear: the step-finding output appears to show *three* bursts, but the main text refers to the figure as showing "two rapid bursts", and the annotation and rate calculation from the text implicitly treat the relaxation as a single burst – 10 passages in 10 s for a rate of 1/s – while the burst rates (that would be extracted for tabulation in Figure 4D) from this trace must be somewhat higher than that (and higher than average, based on 4D), since much of that 10 s period is taken up with pauses; correct?

I had also requested details of the step-finding procedure – currently described with a quick reference to Seol et al. (2016) – but was perhaps insufficiently specific in this request; what choices were made for the window size, probability threshold, and criteria for false positive rejection? I further suggested validating the analysis of processivity using simulated data as was nicely done and argued for in Seol et al. The authors did not take up this suggestion, but addressed an underlying question by making a statistical argument against a distributive model in their response. A version of this argument could be included in the paper, but I think the probabilities of observing long dwells (in a distributive model) are underestimated in the authors' response, since they are based on the extracted *burst* rates (4D). In a distributive model, the rate that governs the dwell statistics would be equal to the *average* relaxation rate (4C), an unbiased measure obtained independent of the fitting procedure. (A comparison with analysis of simulated data would of course be more complete and convincing, since it also includes biasing effects of the analysis such as missed short events.) Overall, the observations of the authors do seem to support modest processivity, but I think is fair for them to caution in the text that rigorous quantification of the processive behavior will require further work and analysis – as has already been hinted at in the discussion of concentration dependence.

Distributions of crossing angles:

I agree that the analysis of crossing angle distributions previously calculated for supercoiled DNA by Vologodskii and Cozzarelli has strengthened the paper. This analysis relates to comparing positively and negatively supercoiled DNA. Additional nuance could be introduced into the discussion comparing supercoiled DNA with catenanes: the authors note that ~90° angles occur much more frequently in catenanes than in supercoiled DNA, which is certainly true for the supercoiling and catenation densities simulated in the referenced Stone et al. paper. Note however that the distribution depends on supercoiling density (Vologodskii and Cozzarelli 1996, Figure 6); for weakly supercoiled DNA, there is a broader distribution centered closer to 90°. In general, I think it is worth pointing out in the discussion (as acknowledged by the authors in their response) that quantitatively relating substrate preferences to crossing angle selection will require simulations of the various substrates under different conditions, beyond the mean crossing angle calculations and selected previously simulated distributions presented here.

---

## [Author Response]

Reviewer #1:[…] • The authors have specified that error bars refer to standard error of the mean but haven't stated numbers or natures of replicates.

We have provided this information in the revised manuscript.

• It would be helpful to show ATPase activity data for positively supercoiled DNA as well as negative in Figure 4A.

We have included ATPase measurements of Topo VI with positive, negative and relaxed DNA in the revised manuscript (Figure 3—figure supplement 1). The new ATPase data show a ~2-fold increase in ATPase rate in the presence of positively supercoiled in comparison to negatively supercoiled, which is consistent with the single-molecule relaxation data. We have also combined these data with published simulations of crossing angle distributions of supercoiled plasmid DNA to obtain an independent estimate of the preferred crossing angle of topo VI, see below.

• In the Discussion section, page 23 paragraph 2, it says that the near-90° crossing angle for braids could explain the switch from distributivity on plectonemes to processivity on braids by "promoting more stable DNA binding and efficient strand passage." This is speculative and a bit at odds with the binding on supercoils, and further raises the question as to whether there is a way to show binding strength (or unbound/bound fractions) for braids/catenanes in parallel with the data for negative supercoil binding tighter than positive. Please comment.

We agree that measuring the binding to catenated DNA would be a useful addition; however, this experiment is currently technically inaccessible due to the lack of substrate availability. However, on further consideration we have revised the proposed model to provide a less speculative explanation for the slight increase in processivity observed with braided substrates. The slight increase in processivity likely arises from more efficient strand capture and passage of braided DNA, rather than stable binding. The data supports the hypothesis that on braided DNA the rate at which the T-segment is captured and passed increases due to the larger crossing angles and can therefore outcompete the rate at which the enzyme dissociates from the DNA. For the case of supercoiled DNA, with crossing angles that deviate further from the preferred crossing angle, the DNA crossing geometry slows T-segment capture and strand passage to a point where the enzyme is more likely to dissociate from the DNA post strand passage then capture a second T-segment. We have revised the text to reflect this more grounded mechanistic proposal for the increased processivity on braided DNA substrates.

• Page 14 paragraph 1 (in regard to Figure 4D) states that the rate and extent of processive bursts are fairly constant over large concentration changes. However, for the rate plot, there is by eye a notable, fairly steady rate dependence (~1.5x increase for positive, ~2x increase for negative), especially in light of how the changes in Figure 6 Figure Supplement 1 C and D are described. It might be better to admit that the rates do increase slightly, but that since the burst sizes do not increase and because the dwell times decrease to some limit, it could still be concluded that the phenomena result from a single, processive enzyme rather than a concentration increase.

The reviewer raises a valid point and we have amended the description of the data in the revised manuscript to reflect this more nuanced interpretation.

• Page 8, end of first paragraph. Why assume that positive crossover selection occurs following DNA cleavage? Why not just posit that the binding of negative crossovers, while favored, is also mildly inhibitory to strand passage? This would seem very analogous to topo IV. Or perhaps there is some confusion as to when preference occurs that could be cleared up in the writing?

We agree that this perspective is equally as valid and have amended the text to reflect both possibilities.

Reviewer #2:[…] To address my comments about burst analysis, I suggest:1. Showing steps of representative traces, together with overlaid step-finding outputs.

This has been added to figure 4, Panel B.

2. Providing details of the step-finding procedure and subsequent analysis. E.g, the legend indicates that bursts were defined as two or more cycles; at what stage was that criterion imposed? Did step-finding also find smaller bursts (e.g. 1 cycle) and then those were excluded from subsequent analysis? Were the reported burst sizes corrected to account for the detection threshold?

The average burst size was computed based on all of the bursts, including the single-cycle bursts. For the analysis of the processive burst rate, the single cycles were omitted as they occur almost instantaneously, such as seen for supercoil relaxation. This has been clarified in the text.

3. Revisiting the simulation approach that was compelling for validating results in Seol et al. For example, if you simulate a "null hypothesis" of distributive activity with the same average braid relaxation rate measured in your assay, and include realistic noise, then apply the analysis pipeline used for experiments, what comes out?

The reviewer raises a good point regarding the extent to which we can discern true processive bursts from the more rapid binding and action of individual distributive enzymes in the braiding relaxation measurements. The comparison that the reviewer is suggesting between a processive enzyme unlinking the braids versus multiple enzymes unlinking the braid in a perfectly distributive manner, but at the same average rate would be indistinguishable under the vast majority of reasonable assumptions about binding rates and enzymatic unlinking rates. In both cases the dwell-times between individual strand passage events would be exponentially distributed with the same mean time given that the average relaxation rates are the same. Although a direct comparison between the bursts of unlinking cannot distinguish between the two scenarios, we can perhaps get to the same point by considering the dwell times between the rapid bursts and compute the probability of observing these dwell times for a scenario in which the binding and relaxation time from in the bursts corresponds to the binding of individual enzymes, each relaxing a single crossing. As an example we can consider the positive burst rate at low enzyme concentration of ~27 strand passages per min and the average dwell-time between bursts of ~22 s (Figure 4 D and F). At a rate of 27 strand passages per min the average time between strand- passages is ~2.5 s. Assuming exponentially distributed binding times with a mean time of 2.5 s, we can compute the probability of observing a pause of 22s or longer; P(t>=22s) ~0.00015. A similar analysis for the relaxation of negative crossings at the same enzyme concentration (~18 strand passages per minute and a dwell time of ~50 seconds) gives a probability of observing 50 second long pauses between binding events, P(t>=50 s) ~6x10^-8^. A similar analysis can be performed for each average rate and each average pause duration between events, but the probabilities of observing both the average rate and the long pauses between events for a scenario of distributive enzymes binding and performing a single strand passage, never exceeds 0.005.

To address my comments about ingle-crossing stringency, I suggest:1. Report directly on tethers that showed no activity; what was the simulated distribution of crossing angles for those tethers? In particular, how does the probability of sampling the preferred crossing angle compare with tethers for which activity was observed?

Tethers that showed no activity had a narrower separation (2e) between the two DNA molecules, resulting in crossing angle distributions peaked at smaller crossing angles for positive crossings, further away from 90°, and therefore a lower probability of obtaining the preferred crossing angle. To confirm this, we included the simulated single crossing angle distributions at two forces for a representative geometry at which topo VI was not able to unlink the single crossing. The analysis included as Figure 8 supplement 2 illustrates that the crossing angle distribution for single-crossing geometries that were not unlinked by topo VI are peaked at crossing angles 10-15 degrees less than single-crossings that were unlinked, and are ~4-150 -fold less likely to obtain the preferred crossing angle of ~87.6°. This very low probability of obtaining the preferred crossing angle likely results in a T-segment capture rate significantly slower than the G-segment off-rate. The off-rate would then out-compete T-segment capture, which would result in a decrease in the strand passage rate consistent with the precipitous decrease in unlinking rate.

This hypothesis will take substantially more work to test, and would be better addressed through kinetic approaches that could directly measure the G-segment release and capture rates. Nonetheless, we can estimate the G-segment off-rate from the size and rate of the processive bursts (figure 4). The average burst time for positive and negative braids was ~6 seconds, which provides a rough estimate of the G-segment binding time. Single positive crossings were relaxed in 4-10 seconds with an average of 6 seconds, comparable to the G-segment binding time. Negative crossings were relaxed in 7-44 seconds, with an average of 12 seconds. Scaling the measured relaxation rates of positive crossings by the relative probability of obtaining the preferred crossing angle for the non-relaxed crossings results in predicted relaxation times of 227 seconds at 1.5 pN and 76 seconds at 1 pN, and considerably longer for negative crossings. These long waiting times exceed the estimated G-segment binding time by an order of magnitude or more. In a simple kinetic competition model in which T-segment capture competes with G-segment unbinding, the probability of T-segment capture will decrease dramatically and the resulting relaxation rate will decrease quadratically with the decreasing T-segment capture rate. The unlinking rate will decrease both because the capture rate decreases, and because the probability of capturing the T-segment before topo VI releases the G-segment also decreases. The unlinking rate is proportional to the product of the probability and rate of T-segment capture. The combination of the low probability of capturing the T-segment prior to G-segment release, and the slow rate of capturing the T-segment would therefore result in a sharp decrease in the unlinking rate of single-crossings with lower probabilities of capturing the T-segment in the correct geometry for passage. Based on this argument, the observed selective relaxation of a narrow range of imposed crossing angles relatively close to 90° is plausible given the estimated G-segment off rate, but verifying this hypothesis in detail would require substantially more experiments, which is beyond the scope of the current work.

2. How much slower must the rates be on these "no activity detected" tethers for relaxation to not be measurable?

In general, tethers with “no activity detected” were either left for up to 1 hour without a single unlinking event or the tethers did not remain attached to the slide surface for long enough to detect activity. This has been clarified in the text.

3. While not as elegant as the internally controlled left-right comparisons, can raw rates of relaxation be compared between different tethers and related to the simulated probabilities of observing the preferred crossing angle?

The reviewer is correct that the crossing angle can in principle be obtained from the rates of relaxing different single-crossing substrates combined with the simulated DNA crossing angle distributions. As the reviewer points out, comparing relaxation rates among different tethers is less elegant, primarily due to variations in topo VI concentration, even for nominally identical concentrations. The left-right relaxation of individual crossings is by far the most reliable approach to measuring the crossing angle. Nonetheless, to address the question posed by the reviewer, we computed the preferred crossing angle by comparing the relaxation rates of all positive single crossings and all negative single crossings. In practice this was done by comparing each set of n-1 measurements with one measurement for all 14 single-crossing measurements. This results in a total of 364 estimates of the crossing angle (13 measures for each handedness for each of 14 different geometries). The average crossing angle computed for each normalization single-crossing geometry, plotted a function of the imposed crossing angle of the normalizing braid (Author response image 1). The average crossing angle obtained through this approach is 86.1 ± 0.5°, which is consistent with the angle obtained from the more precise left-right comparisons for individual crossings presented in the manuscript.

**Author response image 1. sa2fig1:** Average preferred crossing angle for topo VI obtained from comparisons among all right handed crossings and all left handed crossings for single-crossing relaxation experiments. Each point represents the average crossing angle obtained by comparing the relative relaxation times and simulated DNA crossing angle distributions of 26 single-crossing measurements (13 of each handedness) with measurements from one DNA tether, plotted as a function of the average imposed crossing angle of the tether geometry to which the other measurements were compared. The error bars correspond to the SEM. The average preferred crossing angle computed from these cross-comparisons is 86.1 ± 0.5°.

Additional questions and suggestions:Generally the discussion of crossing angles for different substrates (braids and supercoils at difference forces, catenanes and supercoils in the absence of force) would be quantitatively improved by calculating distributions of crossing angles for these cases in a unified analysis; I recognize this is a significant simulation task and currently you reference some past work to support general claims. At minimum I would reference, consider, and discuss early work presenting distributions of juxtaposition angles, from Vologodskii and Cozzarelli (1994 and 1996), which includes simulations of weakly supercoiled DNA (1996) where the juxtapositions that occur are still broadly distributed and just starting to be asymmetric.

We agree that obtaining the distributions of crossing angles in supercoiled DNA and in braided DNA would permit additional quantitative tests of the crossing angle preference of Topo VI. As the reviewer points out, this is an involved simulation undertaking that has not been previously reported. Our preliminary efforts reveal that this will be a significant undertaking that is beyond the scope or timeframe of this work. However, as suggested by the reviewer, we considered the crossing angle distributions for supercoiled DNA obtained by Vologodskii and Cozarelli in their 1994 and 1996 publications. Based on these published distributions estimated the preferred crossing angle based on the relative topo VI relaxation rate (more precisely the ATPase rate) of negatively versus positively supercoiled plasmid DNA (Figure 3 Figure Supplement 1). We find that for the preferred crossing angle of 87.6 deg, the relative probabilities in positively verses negatively supercoiled plasmids (both positive and negative σ = 0.06) are ~0.7, in reasonable agreement with the measured value of ~0.5. Alternatively, the ratio of crossing angle probabilities is a factor of ~0.5 at a crossing angle of ~85 degrees. Intriguingly, under the assumption that the positively supercoiled DNA is slightly (~15%) less supercoiled (σ = 0.05) that the negatively supercoiled (σ = -0.06) DNA, the agreement between the measured relaxation rate difference and relative probability is near perfect. Given the uncertainties in these measurements and the precise level of negative versus positive supercoiling in the plasmids, the correspondence is reasonable and is consistent with the predicted crossing angles measured with much higher precision in the single-crossing experiments.

We thank the reviewer for making this insightful suggestion. We have included this analysis in the revised manuscript as Figure 8 Figure Supplement 1.

The biochemical decatenation assay used here and in Ref. 47 is quite striking. What can you say about how "special" topo VI is in terms of the results of that assay? Ref. 47 shows a similar result for topo II α, where catenation occurs at a lower enzyme concentration than supercoil relaxation. How about other topo II enzymes, e.g. yeast topo II, bacterial topo IV…?

In the paper mentioned (ref 47) they also look at gyrase decatenation which appears to require more enzyme that topo IIa. However, as this substrate is very new in the field, data is not currently available to comment on concerning the other type II topoisomerases. However, hopefully in the near future it will be applied more broadly and any differences in the ways the type II topoisomerases catalyse the decatenation of this substrate will be revealed.

[Editors' note: further revisions were suggested prior to acceptance, as described below.]

The manuscript has been much improved. Both referees are favorable but request that a couple of issues still be addressed, as outlined below:Reviewer #1:The authors have satisfactorily addressed all major comments. A couple typos still appear to have snuck through: "WaveMentrics" in Figure 1C caption, and the formatting of Weeks-Chandler potential is fixed but the Debye-Huckle is not.

Thank you, these have now been corrected.

Reviewer #2:In their revision, McKie et al. have added substantial new analyses, experiments, and clarifications to their manuscript, strengthening their study while responding thoughtfully to many of the issues raised by both reviewers. I am supportive of publication in eLife, and my remaining comments are concerned with just a few points on which I feel the authors could be more complete in addressing questions from the previous round of review, without requiring additional experiments or extensive calculations.Burst analysis for unbraiding analysis:I previously asked that the authors show representative traces, together with overlaid step-finding outputs. This request was minimally addressed by adding the step-finding overlay to the single trace that was already included in Figure 4B. To give the reader a more representative sense of this data and analysis, I think it would be appropriate to show more traces and overlays, including under varying conditions; this could be done in a supplementary figure. The annotation and description of Figure 4B itself could also be more clear: the step-finding output appears to show three bursts, but the main text refers to the figure as showing "two rapid bursts", and the annotation and rate calculation from the text implicitly treat the relaxation as a single burst – 10 passages in 10 s for a rate of 1/s – while the burst rates (that would be extracted for tabulation in Figure 4D) from this trace must be somewhat higher than that (and higher than average, based on 4D), since much of that 10 s period is taken up with pauses; correct?

The reviewer raises a good point and we apologize for not providing a more thorough description of the step-finder analysis of the braided DNA relaxation data in the previous version of the manuscript. We have included an additional supplementary figure (Figure 4 Figure Supplement 1) showing the measured extension data and step-finding fit overlays for several representative braid relaxation events under various conditions.

In regards to figure 4B, we thank the reviewer for noticing the errors in reporting the number of processive bursts in the example trace, in addition to the confusing language in the text and the figure legend in describing the relaxation events. As the reviewer points out, there are three bursts, not two. With respect to the relaxation rate, we reported the average rate corresponding to the beginning and ending of the relaxation as indicated by the red dotted line in Figure 4B, which is a potential source of confusion since the overall average relaxation rate between the introduction of the braid and its relaxation was about half that at ~0.5 reactions s^-1^ or 30 Lk min^-1^. We have revised the main text and figure legend to correct these errors and confusing language. The reviewer is correct that the burst rates observed in the example trace in Figure 4B are above the mean. Individual burst rates were widely distributed, which is made clear in the additional supplemental figure exhibiting a number of representative braid relaxation traces. The burst and average relaxation rates displayed in Figure 4B are higher than the average, but are nonetheless representative of the broad distribution of average and burst relaxation rates.

I had also requested details of the step-finding procedure – currently described with a quick reference to Seol et al. (2016) – but was perhaps insufficiently specific in this request; what choices were made for the window size, probability threshold, and criteria for false positive rejection? I further suggested validating the analysis of processivity using simulated data as was nicely done and argued for in Seol et al. The authors did not take up this suggestion, but addressed an underlying question by making a statistical argument against a distributive model in their response. A version of this argument could be included in the paper, but I think the probabilities of observing long dwells (in a distributive model) are underestimated in the authors' response, since they are based on the extracted burst rates (4D). In a distributive model, the rate that governs the dwell statistics would be equal to the average relaxation rate (4C), an unbiased measure obtained independent of the fitting procedure. (A comparison with analysis of simulated data would of course be more complete and convincing, since it also includes biasing effects of the analysis such as missed short events.) Overall, the observations of the authors do seem to support modest processivity, but I think is fair for them to caution in the text that rigorous quantification of the processive behavior will require further work and analysis – as has already been hinted at in the discussion of concentration dependence.

We thank the reviewer for clarifying their concerns with the braid unlinking data analysis and interpretation. We also appreciate the good suggestion of simulating the braid unlinking as a series of distributive relaxation events occurring at the global average relaxation rate to compare the t-test analysis of these simulated traces with the results of the t-test analysis of the experimental braid unlinking data. We have added supplements to figure 4 (Figure supplements 2 and 3) in which we simulate a range of unlinking rates as a series of single relaxation steps occurring at exponentially distributed times corresponding to the average relaxation rate. We compute the average relaxation step-size, the average pause duration, and the probability of a processive burst (relaxation steps larger than 150% of a single crossing) for the positive, negative, and simulated braid relaxation data analyzed with identical t-test parameters. The results lend additional support to the conclusion that braided DNA is unlinked in a series of short processive bursts that are separated by long dwell times associated with topo VI rebinding. As is clear from the comparison between the simulated and measured T-test analysis results, the experimental average step-sizes, average dwell times, and probabilities of processive bursts are substantially larger than the values from the simulations based on the assumption of purely distributive relaxation events occurring at the same average rate. Whereas the T-test analysis does occasionally miss individual steps thereby artifactually indicating processive steps, the probability and extent of these false positive events are much lower than observed in the t-test analysis of the experimental braid relaxation data. We have also indicated the t-test parameters used for the analysis in the supplement. The data was initially down-sampled from 200 Hz to 20 Hz. The T-test comparison window size was set to 40 points, the significance level (α parameter) was set to an extremely stringent value of 10^-7^, and the minimum step size was set to half the extension associated with a single braid relaxation (typically 40 nm).

Distributions of crossing angles:I agree that the analysis of crossing angle distributions previously calculated for supercoiled DNA by Vologodskii and Cozzarelli has strengthened the paper. This analysis relates to comparing positively and negatively supercoiled DNA. Additional nuance could be introduced into the discussion comparing supercoiled DNA with catenanes: the authors note that ~90° angles occur much more frequently in catenanes than in supercoiled DNA, which is certainly true for the supercoiling and catenation densities simulated in the referenced Stone et al. paper. Note however that the distribution depends on supercoiling density (Vologodskii and Cozzarelli 1996, Figure 6); for weakly supercoiled DNA, there is a broader distribution centered closer to 90°. In general, I think it is worth pointing out in the discussion (as acknowledged by the authors in their response) that quantitatively relating substrate preferences to crossing angle selection will require simulations of the various substrates under different conditions, beyond the mean crossing angle calculations and selected previously simulated distributions presented here.

We thank the reviewer for pointing out this consideration related to the dependence of the crossing angle of catenated DNA on the level of supercoiling of the linked DNA molecules. We have included this additional consideration along with a citation to the Vologodskii and Cozzarelli paper in the Discussion section (p 18) of the revised manuscript.